# IDENTITY-FREE DEFERRAL FOR UNSEEN EXPERTS

**Joshua Strong,** * **Pramit Saha, Yasin Ibrahim, Cheng Ouyang, J. Alison Noble**
Department of Engineering Science, University of Oxford, United Kingdom

## ABSTRACT

Learning to Defer (L2D) improves AI reliability in decision-critical environments by training AI to either make its own prediction or defer the decision to a human expert. A key challenge is adapting to unseen experts at test time, whose competence can differ from the training population. Current methods for this task, however, can falter when unseen experts are out-of-distribution (OOD) relative to the training population. We identify a core architectural flaw as the cause: they learn identity-conditioned policies by processing class-indexed signals in fixed coordinates, creating shortcuts that violate the problem's inherent permutation symmetry. We introduce Identity-Free Deferral (IFD), an architecture that enforces this symmetry by construction. From a few-shot context, IFD builds a query-independent Bayesian competence profile for each expert. It then supplies the deferral rejector with a low-dimensional, role-indexed state containing only structural information, such as the model's confidence in its top-ranked class and the expert's estimated skill for that same role, which obscures absolute class identities. We train IFD using an uncertainty-aware, context-only objective that removes the need for expensive query-time expert labels. We formally prove the permutation invariance of our approach, contrasting it with the generic non-invariance of standard population encoders. Experiments on medical imaging benchmarks and ImageNet-16H with real human annotators show that IFD consistently improves generalisation to unseen experts, with gains in OOD settings, all while using fewer annotations than alternative methods.

## 1 INTRODUCTION

In high-stakes domains such as healthcare, autonomous systems must decide not only *what* to predict but also *when* to defer to a human. *Learning to Defer* (L2D) formalises this by training a classifier to predict labels and a rejector to decide whether to accept the AI's prediction or defer to an expert, directly optimising end-to-end performance of the resulting human-AI collaborative system (Madras et al., 2018). This design promises safer AI deployment, but real-world environments pose a key challenge: the experts available at test time are often *unseen* (i.e., not among the experts observed during training). Hospitals rotate staff, trainees progress, and practice varies across sites. As such, a robust deferral system must adapt to such turnover.

Existing population-adaptive extensions of L2D tackle this challenge by meta-learning expert representations from a small *context* of past predictions (Tailor et al., 2024). Conditioning the rejector on these embeddings enables transfer to unseen experts drawn from the same distribution (i.e., in-distribution (ID)). However, in practice, systems must also handle *out-of-distribution* (OOD) unseen experts whose competence profiles are systematically shifted—for example, the same specialisation pattern under a different class indexing or altered strengths across sites. Here, existing methods can falter. We diagnose the root cause: architectural *identity-conditioned deferral*. By processing class-indexed signals in fixed coordinates (e.g., per-class embeddings), population encoders expose label identities. This allows the rejector to learn shortcuts such as "if index 3 is strong, defer," which may work for the training label indexing but break under simple class relabellings (i.e., re-naming/reindexing classes should not change the deferral decision). As a result, such policies can misgeneralise under reindexings common in OOD settings.

**Our approach.** We propose *Identity-Free Deferral* (IFD), which enforces the correct inductive bias *architecturally*. IFD eliminates all absolute identity channels before the rejector sees any input.

---

*Correspondence:`joshua.strong@eng.ox.ac.uk`

From a few-shot context, it builds a Bayesian competence profile for each class. IFD then supplies the rejector with a role-indexed state: values read at structural roles such as the model's top class or the expert's estimated best class, plus symmetric aggregates. This low-dimensional interface ensures that deferral decisions depend only on transferable structure, never on class IDs. Training uses an uncertainty-aware, context-only surrogate objective, which naturally downweights uncertain profiles and eliminates the need for query-time expert labels.

**Why this transfers.** By aligning the architecture with the permutation symmetry of class relabellings, IFD blocks identity-conditioned shortcuts. Its query-independent profiles reduce variance at small context sizes and scale gracefully as more context is available. The resulting policy focuses on the structural question—*who is better for this case?*—rather than memorising identities.

In summary, our contributions are:

- **Identity-Free Deferral (IFD):** a role-indexed rejector interface that enforces invariance to class relabelling by construction. We formally prove this property and show it is not guaranteed by standard population encoders.
- **Context-only, uncertainty-aware training:** a data-efficient objective for adapting to new experts without query-time expert labels, enabled by explicit Bayesian competence profiles that naturally support uncertainty quantification and the incorporation of prior knowledge.
- **Empirical findings:** across medical and general imaging benchmarking datasets, IFD improves generalisation to unseen experts, with the largest gains under OOD expert shifts.

## 2 RELATED WORK

An extended survey on related work appears in App. A.

**Learning to Defer (L2D).** L2D augments selective prediction/rejection (Chow, 1957; 1970; Cortes et al., 2016) by training a classifier and a rejector that can defer to an expert, improving end-to-end outcomes over prediction-only systems (Madras et al., 2018; Mozannar & Sontag, 2021). Follow-ups study multiple experts, limited expert signals, cost/triage constraints, and two-stage training (Verma et al., 2023; Hemmer et al., 2023; Alves et al., 2024; Mao et al., 2023; Montreuil et al., 2025b), with applications in healthcare, fraud, and content moderation (Joshi et al., 2021; Strong et al., 2025; Alves et al., 2023; Lykouris & Weng, 2024).

**Adapting to a population of experts.** The state-of-the-art approach for handling unseen experts is L2D-Pop (Tailor et al., 2024), which meta-learns an expert representation from a few-shot *context* to condition the rejector. While effective for ID experts, its high-capacity encoders process class information in fixed coordinates, allowing the model to learn the identity-conditioned shortcuts we identify as a key failure mode. These architectures lack a sufficient inductive bias for permutation symmetry, which hinders generalisation to OOD experts.

Our work, *Identity-Free Deferral (IFD)*, addresses this architectural gap directly. Instead of learning an unstructured latent embedding, IFD constructs an explicit, query-independent Bayesian profile of an expert's competence. Crucially, it communicates this information to the rejector through a low-dimensional, *role-indexed* state that is provably invariant to class relabelling. This design is complemented by a context-only, uncertainty-aware loss that removes the need for expensive query-time labels, further improving data efficiency.

## 3 BACKGROUND & PROBLEM SETUP

### 3.1 STANDARD (SINGLE EXPERT) L2D

In a classification task, we predict a target $Y \in \mathcal{Y} \equiv \{1, \ldots, K\}$ from inputs $X \in \mathcal{X}$, where $(X, Y) \sim \mathcal{P}$. The L2D framework assumes an expert $E$ provides predictions $M \in \mathcal{M} = \mathcal{Y}$. Given a dataset $\mathcal{D} = \{(x_i, y_i, m_i)\}_{i=1}^{N}$, the objective is to learn a predictor $\hat{Y} : \mathcal{X} \to \mathcal{Y} \cup \{\bot\}$ ($\bot$ denotes deferral) by jointly training a *classifier* $h : \mathcal{X} \to \mathcal{Y}$ and a *rejector* $r : \mathcal{X} \to \{0, 1\}$. $h$ predicts the target; $r$ decides to use $h(x)$ ($r(x) = 0$) or defer ($r(x) = 1$). For this subsection only, the expert is fixed (i.e., a single expert $E_0$ supplies all responses $M$ at both train and test time – no other experts can enter or leave the system) so we write $r(x) \equiv r(x, E_0)$ for brevity.

**Optimisation.** Assuming costs $l(x, y, h(x))$ for classification and $l_{\exp}(x, y, m)$ for deferral, the system-level L2D loss is:

$$\mathcal{L}(h, r) = \mathbb{E}_{(x,y)\sim\mathcal{P}, \, m\sim P(\cdot|x,y,E_0)}\Big[l(x,y,h(x))\,\mathbb{I}\{r(x)=0\} \; + \; l_{\exp}(x,y,m)\,\mathbb{I}\{r(x)=1\}\Big]. \quad (1)$$

This non-convex loss is hard to optimise directly; Mozannar & Sontag (2021) proposed a consistent surrogate for Eq. (1), derived from the Bayes minimiser of the 0–1 system loss. The 0–1 loss is:

$$\mathcal{L}_{0-1}(h, r) = \mathbb{E}_{(x,y)\sim\mathcal{P}, \, m\sim P(\cdot|x,y,E_0)}[\mathbb{I}\{h(x)\neq y\}\,\mathbb{I}\{r(x)=0\} \; + \; \mathbb{I}\{m\neq y\}\,\mathbb{I}\{r(x)=1\}].$$

Minimising $\mathcal{L}_{0-1}$ yields the Bayes-optimal classifier $h^*(x)$ and rejector $r^*(x)$:

$$h^*(x) := \arg\max_{y\in\mathcal{Y}} P(Y{=}y \mid x), \quad r^*(x) := \mathbb{I}\{\, P(Y{=}M \mid x, E_0) \geq \max_{y\in\mathcal{Y}} P(Y{=}y \mid x)\,\}. \quad (2)$$

With score functions $g_y(x)$ for $h(x) = \arg\max_y g_y(x)$ and $g_\perp(x)$ for $r(x) = \mathbb{I}\{\max_y g_y(x) \leq g_\perp(x)\}$, a consistent softmax-based surrogate is:

$$\mathcal{L}_{CE}(g_1, \ldots, g_K, g_\perp; x, y, m) := -\log p(y \mid x) \; - \; \mathbb{I}\{m = y\}\log p(\perp\mid x), \quad (3)$$

where $p(y' \mid \cdot) = \exp\{g_{y'}(\cdot)\}/\sum_{y''\in\mathcal{Y}\cup\{\perp\}} \exp\{g_{y''}(\cdot)\}$.

This loss encourages accurate classification (via $g_y$) and deferral (via $g_\perp$) when the expert is correct ($m = y$). At inference, prediction is deferred if $g_\perp(x) \geq \max_{y\in\mathcal{Y}} g_y(x)$. Standard L2D assumes a single, fixed expert. As a result, the learned rejector is tailored to that expert's behaviour; substituting a different (unseen) expert at test time is unsupported here. This limitation motivates population-adaptive L2D, described next.

## 3.2 POPULATION-ADAPTIVE L2D

The standard approach to the problem of fixed-expert coupling is to adapt the *rejector* to the identity of the test-time expert $E$ using a small *context set* $D_C^E = \{(x_i^C, y_i^C, m_i^{E,C})\}_{i=1}^{N_C^E}$ summarising that expert's past behaviour. Here, the test-time expert varies, so we henceforth denote the rejector and its respective score function conditioned on $E$ as $r(x, E)$ and $g_\perp(x, E)$, respectively.

Tailor et al. (2024) propose learning a set encoder $\psi$ which maps context $D_C^E$ to a fixed-$d_\psi$ dimensional latent expert representation. There are two variants:

**Query-independent (QI):** $\quad \psi^E = \psi(D_C^E) \in \mathbb{R}^{d_\psi}, \; \text{with } g_\perp(x, E) = g_\perp(x, \psi^E), \quad (4)$

**Query-conditioned (QC):** $\quad \psi^E(x) = \psi(x, D_C^E) \in \mathbb{R}^{d_\psi}, \; \text{with } g_\perp(x, E) = g_\perp(x, \psi^E(x)). \quad (5)$

The QI variant uses a Conditional Neural Process (Garnelo et al., 2018), the QC variant an Attentive Neural Process (Kim et al., 2019b). Note that both encode the context: $[\varphi(x_i^C); \, e(y_i^C); \, e(m_i^{E,C})]$, where $e(\cdot)$ is a one-hot/learned label embedding and $\varphi$ a feature encoder.

**Bayes rule.** Averaging the system 0–1 loss over the expert population $E \sim \mathcal{Q}$ yields

$$\mathcal{L}_{0-1}^{\text{pop}}(h, r) = \mathbb{E}_{(x,y)\sim\mathcal{P}, \, E\sim\mathcal{Q}, \, m\sim P(\cdot|x,y,E)}[\mathbb{I}\{r(x,E)=0\}\,\mathbb{I}\{h(x)\neq y\} \; + \; \mathbb{I}\{r(x,E)=1\}\,\mathbb{I}\{m\neq y\}].$$

The classifier is unchanged, and the rejector compares the expert-conditioned correctness to the model's best class:

$$h^*(x) = \arg\max_{y\in\mathcal{Y}} P(Y{=}y \mid x), \; r^*(x, E) = \mathbb{I}\{\, P(Y{=}M \mid x, E) \geq \max_{y\in\mathcal{Y}} P(Y{=}y \mid x)\,\}. \quad (6)$$

**Surrogate and training.** Let

$$p(y \mid x, \psi) = \frac{\exp\{g_y(x)\}}{Z(x,\psi)} \quad \text{for } y \in \mathcal{Y}, \qquad p(\perp\mid x, \psi) = \frac{\exp\{g_\perp(x,\psi)\}}{Z(x,\psi)},$$

where $Z(x,\psi) := \exp\{g_\perp(x,\psi)\} + \sum_{y\in\mathcal{Y}} \exp\{g_y(x)\}$. A consistent surrogate follows Eq. (3):

$$\mathcal{L}_{CE}^{\text{pop}}(g_1, \ldots, g_K, g_\perp; \, x, y, m, \psi) = -\log p(y \mid x, \psi) \; - \; \mathbb{I}\{m{=}y\}\log p(\perp\mid x, \psi). \quad (7)$$

Inference is analogous: predict $h(x) = \arg\max_y g_y(x)$ and defer iff $g_\perp(x, \psi) \geq \max_y g_y(x)$.

**Multi-expert extension.** Given an available set of experts $\{E_j\}_{j=1}^J$, we can evaluate $g_\perp(x, E_j)$ for each $j$ and defer to the expert with the largest deferral margin. See Remark 4.1 for the explicit rule and its relation to the standard $K{+}J$-way surrogate (Verma et al., 2023).

### 3.3 PERMUTATION SYMMETRY, IDENTITY LEAKAGE, AND UNSEEN VS. OOD EXPERTS

In this subsection we make explicit the permutation symmetry that the population problem enjoys and explain why architectural violations of that symmetry predict poor transfer to OOD experts. Proving (non-)invariance under *coherent relabellings* (CR) is an architectural, distribution-free property: it reveals whether the hypothesis class permits dependence on absolute label identities. This serves as a clean proxy for vulnerability to the expert-only reindexings used in our OOD stress tests. The CR non-invariance result establishes the possibility of identity-conditioned shortcuts under standard parameterisations; empirical failures then arise from standard ERM and optimisation dynamics, which select such shortcuts whenever they reduce empirical risk.

**The core symmetry: coherent relabelling.** For any class-indexed vector $v = (v_y)_{y \in \mathcal{Y}}$ and permutation $\pi \in \mathfrak{S}_K$, define the permuted vector $v^\pi$ by $(v^\pi)_y := v_{\pi^{-1}(y)}$. A *coherent relabelling* applies the same permutation to every class-indexed object (model outputs, labels, expert profiles, etc.). If the original data mechanism is $P(Y \mid X)$, the relabelled mechanism is

$$P^\pi(Y{=}y \mid X{=}x) := P(Y{=}\pi^{-1}(y) \mid X{=}x).$$

Write the population Bayes action from (6) as

$$f_{\text{pop}}^\star(x, E; P, M^E) = \big(h^\star(x; P),\, r^\star(x, E; P, M^E)\big).$$

**Proposition 3.1** (Classifier equivariance and rejector invariance under coherent relabelling)**.** *For any $\pi \in \mathfrak{S}_K$, we have:*

1. **Classifier equivariance**: $h^\star(x; P^\pi) = \pi(h^\star(x; P))$.
2. **Rejector invariance:** $r^\star(x, E; P^\pi, M^{E,\pi}) = r^\star(x, E; P, M^E)$.

*Proof sketch.* Classifier equivariance is immediate: applying $\pi$ simply permutes the label space. For the rejector, all quantities that determine the Bayes decision (e.g. model top probability and the expert's classwise correctness terms used in the expected expert accuracy) are permuted coherently and therefore remain identical as real numbers. Hence the binary defer/predict decision is unchanged.

**ID-unseen vs. OOD-unseen experts and the expert-only stress test.** An expert is *ID-unseen* if it is new at test time but drawn from the same population as the training experts; an *OOD-unseen* expert has a competence pattern outside this population. To model a simple, interpretable OOD shift we consider *expert-only permutations*: the task and model posterior remain fixed, but the expert competence vector is reindexed.

Let $\mu_y^E(x) := P(M^E{=}y \mid X{=}x, Y{=}y, E)$ be the expert's class-conditional correctness at $x$, and let $\eta_y(x) := P(Y{=}y \mid X{=}x)$. Define the expected expert correctness

$$q(x, E) = \sum_{y \in \mathcal{Y}} \eta_y(x)\, \mu_y^E(x).$$

An expert-only permutation yields $(\mu^{E,\pi})_y(x) = \mu_{\pi^{-1}(y)}^E(x)$, giving

$$q(x, E^\pi) = \sum_y \eta_y(x)\, \mu_{\pi^{-1}(y)}^E(x) = \sum_z \eta_{\pi(z)}(x)\, \mu_z^E(x).$$

As the permutation alters how expert competence is reweighted by model probabilities, $q(x, E^\pi)$ can differ from $q(x, E)$ and may change the Bayes decision (as is to be expected).

**Proposition 3.2** (Expert-only permutations can change the Bayes action)**.** *If $q(x, E^\pi) \neq q(x, E)$ and the deferral margin is nondegenerate at $(x, E)$, then $r^\star(x, E^\pi) \neq r^\star(x, E)$.*

*Proof sketch.* Let $\Delta_{\text{sys}}(x, E) := q(x, E) - \max_y \eta_y(x)$. The Bayes rule defers iff $\Delta_{\text{sys}}(x, E) \geq 0$. Any permutation that flips the sign of this margin (achievable by aligning large $\mu_z^E$ with small or large $\eta_{\pi(z)}$ appropriately) flips the Bayes defer/predict decision when the margin is nondegenerate.

**Where architectures can fail: identity channels.** Architectures that expose absolute class identities (e.g. one-hots/untied embeddings/coordinate-specific per-class channels) allow rejectors to form rules that depend on fixed coordinates of class vectors. These *label-identity channels* enable index-tied shortcuts such as "defer when coordinate $j$ is strong," which interpolate well in-distribution but are brittle under expert-only permutations.

**Theorem 3.1** (Generic non-invariance of standard L2D-Pop parameterisations). *Under standard untied class-indexed parameterisations (explicit one-hots or untied embeddings feeding per-class channels), both the QI and QC population encoders generically fail to be invariant to coherent relabellings: there exist $(x, E, \pi)$ such that $g_\perp(x, E^\pi) \neq g_\perp(x, E)$.[1] See App. E.2–E.3 for formal statements and proofs.*

*Proof sketch.* QI and QC encoders consume class-indexed inputs in fixed coordinates and parameterise each coordinate with untied weights. As a result, the network can attach semantic meaning to absolute indices (e.g. treating coordinate 3 as "the expert's strong class"). Under a CR $\pi$, the expert's strongest class moves from coordinate $j$ to $\pi(j)$, but the encoder's weights tied to coordinate $j$ do not move with it. Thus the latent representation typically changes in a way that does not correspond to the relabelling. Formally, Appendix E.2–E.3 show that, for a full-measure set of parameters, there exist $(x, E, \pi)$ with $g_\perp(x, E^\pi) \neq g_\perp(x, E)$. Achieving invariance would require non-generic weight-tying patterns that these architectures do not impose.

**Corollary 3.1** (Architectural vulnerability to expert-only permutations). *If an encoder/rejector is non-invariant to coherent relabellings as in Thm. 3.1, then there exist inputs $x$, expert profiles $E$, and expert-only permutations $\pi$ for which the rejector output changes when only the expert profile is permuted. In short: architectural non-invariance to the problem's symmetry implies vulnerability to expert-only permutations.*

**Remarks.** Non-invariance is an *existential* architectural diagnosis: it shows that absolute-index shortcuts are representable. Whether training actually selects such shortcuts depends on the data distribution and the optimiser but is routinely observed in practice. Conversely, architectures that enforce CR-invariance eliminate these shortcuts by construction.

**Lay summary.** The population Bayes rule respects coherent relabelling (Prop. 3.1). Standard L2D-Pop architectures break this symmetry via identity-indexed channels and are therefore generically non-invariant (Thm. 3.1), implying vulnerability to expert-only permutations (Cor. 3.1). This motivates our OOD stress tests and, in §4, our Identity-Free Deferral architecture, which enforces the relevant symmetry by construction.

## 4 METHOD: IDENTITY-FREE DEFERRAL (IFD)

Motivated by the symmetry analysis in §3.3, we propose *Identity-Free Deferral* (IFD): a deferral mechanism that enforces the problem's permutation symmetry *architecturally*. IFD removes all label/expert identity channels *before* the rejector sees any input, so the rejector can depend only on *structural* relations between the model's beliefs and an expert's competence. Consequently, the learned policy is insensitive to coherent relabellings (matching the Bayes-rule equivariance), avoids identity-conditioned shortcuts characteristic of standard population encoders, and transfers to both *ID- and OOD-unseen* experts whose competence profiles are permuted or shifted relative to training.

**Overview.** From a few-shot context, we form a query-independent Bayesian competence profile per class (e.g., Beta–Binomial posterior mean/variance) plus a Lower Confidence Bound (LCB) weight encoding uncertainty/priors (§4.1). We then build an identity-free, role-based state with a permutation-invariant rejector (§4.2) and map it to a deferral logit via a small MLP. Finally, we train with an uncertainty-aware cross-entropy that uses only these profiles and LCBs—no query-time expert labels—and accommodates prior knowledge (§4.3, §4.4). We follow the notation of §3.

### 4.1 FROM CONTEXT TO AN EXPLICIT BAYESIAN EXPERT PROFILE

The purpose of this subsection is to describe how, from a few-shot context of an expert's past predictions, we compute an explicit Bayesian per-class competence profile. Importantly, because the role-indexed state depends only on values read at structural roles (e.g., model-top and expert-top) and on permutation-invariant aggregates, it removes all label-identity channels. This enforces the coherent-relabelling symmetry identified in §3.3 and blocks the generic non-invariance mechanisms that standard population encoders permit.

---

[1]"Generically" means on an open (full-measure) set of parameter values; achieving invariance would require explicit weight tying or symmetry constraints.

We model per-class correctness via Beta–Binomial conjugacy. For each class $y \in \mathcal{Y}$, let $n_y^E = \sum_i \mathbb{I}\{y_i^C = y\}$, $t_y^E = \sum_i \mathbb{I}\{y_i^C = y, m_i^{E,C} = y\}$. With prior $\theta_y^E \sim \text{Beta}(\alpha_y, \beta_y)$, the posterior is:

$$\theta_y^E \mid D_C^E \sim \text{Beta}(\alpha_y + t_y^E, \ \beta_y + n_y^E - t_y^E).$$

Two important properties of the posterior are the classwise mean and variance: $\mu_y^E := \mathbb{E}[\theta_y^E \mid D_C^E]$, $(\sigma_y^E)^2 := \text{Var}[\theta_y^E \mid D_C^E]$. We identify the estimated strongest class $\widehat{k}_{\text{best}}^E := \arg\max_y \mu_y^E$.

Note that we approximate $P(M^E = y \mid Y = y, E)$ rather than $P(M^E = y \mid X = x, Y = y, E)$ to improve data efficiency and reduce spurious instance-level couplings. In high-stakes domains such as medical imaging, inter-expert variation is often aligned with class-level specialisation. The class-level abstraction yields (i) closed-form posteriors and uncertainty; (ii) a state that transfers across experts; and (iii) a natural handle for prior knowledge (§4.4). Because our method's deferral logic depends on knowing the expert's strongest class, we must ensure this class can be identified reliably from a finite context, as Proposition 4.1 guarantees.

**Proposition 4.1** (Peak-class identification). *Let an arbitrary expert have true but unknown per-class accuracies $\bar{\theta}_k \in [0, 1]$ (we drop the $E$ notation here). Let $k_{best} = \arg\max_k \bar{\theta}_k$ be the unique class of maximal expertise. As the number of context samples $n_k \to \infty$ for each class $k$, the posterior mean estimates $\mu_k$ converge almost surely to $\bar{\theta}_k$, and the selected expertise class $\arg\max_k \mu_k$ converges almost surely to $k_{best}$. Moreover, if each class $k$ is observed at least $n$ times, where*

$$n \geq \frac{\ln\left(\frac{2K}{\delta}\right)}{2\left(\frac{\Delta_{acc}}{2}\right)^2}, \quad \text{with } \Delta_{acc} := \bar{\theta}_{k_{best}} - \max_{k \neq k_{best}} \bar{\theta}_k.$$

*then with probability at least $1 - \delta$, the posterior means satisfy $\mu_{k_{best}} > \mu_k$ for all $k \neq k_{best}$; that is, the correct expertise class is identified. Proof: See App. D.1.*

### 4.2 IDENTITY-FREE, ROLE-INDEXED STATE AND REJECTOR

In this subsection we show how the Bayesian per-class profile of §4.1 and the classifier's scores are converted into a compact, role-indexed state (values read at structural roles like the model-top and expert-top plus symmetric aggregates). By exposing only these identity-free features to the rejector, we block label-identity channels and guarantee the rejector's decisions are permutation-invariant and transferable across unseen experts.

Let the classifier produce $\rho(x) = \text{softmax}(g(x))$ with coordinates $\rho_k(x)$ and $k_{\text{top}}(x) \in \arg\max_k \rho_k(x)$. From the expert's context $D_C^E$, fix $J_{\mathcal{F}} \geq 1$ per-class posterior functionals $\mathcal{F} = \{f_j\}_{j=1}^J$ (e.g., mean, variance, MAP, quantiles/LCBs, log-odds, updated hyperparameters). Define

$$f_j^E(y) := f_j(\theta_y^E \mid D_C^E), \qquad \Phi^E := [\, f_j^E(y)\,]_{y=1..K, \ j=1..J_{\mathcal{F}}} \in \mathbb{R}^{K \times J_{\mathcal{F}}}.$$

Choose a ranking functional $u \in \mathcal{F}$. For any $v \in \mathbb{R}^K$, let $v_{(1)} \geq v_{(2)} \geq \cdots \geq v_{(K)}$ denote order statistics (with ties broken by index).

**Identity-free state.** The IFD rejector never sees absolute class IDs. Instead, it receives a small vector $z(x, E) \in \mathbb{R}^{d_z}$ made only of (i) values *read at roles*—indices picked by the data, not by name—and (ii) permutation-invariant aggregates. The two roles we use are the predictor's top class $k_{\text{top}}(x)$ and the expert's top class from context $\widehat{k}_{\text{best}}^E$. Define $z(x, E) = \text{RI}(\rho(x), \Phi^E)$, where RI returns a fixed set of features such as $\rho_{k_{\text{top}}(x)}(x)$, $u_{k_{\text{top}}(x)}^E$, $\rho_{\widehat{k}_{\text{best}}^E}(x)$, $u_{\widehat{k}_{\text{best}}^E}^E$, the indicator $\mathbb{I}\{k_{\text{top}}(x) = \widehat{k}_{\text{best}}^E\}$, the gaps $\rho_{k_{\text{top}}(x)}(x) - \rho_{(2)}(x)$ and $u_{(1)}^E - u_{(2)}^E$, and symmetric aggregates of $\rho$ (e.g., entropy). As entries are either read at roles or are symmetric aggregates, $z(x, E)$ does not depend on absolute class identities.

**Our instantiated feature set (used in all experiments).** We take $J=2$ posterior functionals (per-class mean and variance) and rank by mean $u_y^E \equiv \mu_y^E$. Our 6-dimensional state is

$$z(x, E) = \Big( \underbrace{\rho_{k_{\text{top}}(x)}(x), \ \rho_{\widehat{k}_{\text{best}}^E}(x)}_{\text{model confidences at roles}}, \ \underbrace{\mu_{k_{\text{top}}(x)}^E, \ (\sigma_{k_{\text{top}}(x)}^E)^2}_{\text{expert profile at model-top}}, \ \underbrace{\mu_{\widehat{k}_{\text{best}}^E}^E, \ (\sigma_{\widehat{k}_{\text{best}}^E}^E)^2}_{\text{expert profile at expert-top}} \Big) \in \mathbb{R}^6. \qquad (8)$$

**Rejector.** An MLP maps the state to the deferral logit: $g_\perp(x, E) := r_\phi\big(z(x, E)\big), r_\phi : \mathbb{R}^{d_z} \to \mathbb{R}$.

**Permutation invariance (why this blocks identity leakage).** If a permutation $\pi \in \mathfrak{S}_K$ reindexes all class-indexed quantities coherently, $(\rho^\pi)_y = \rho_{\pi^{-1}(y)}$ and $(\Phi^{E,\pi})_{y,j} = (\Phi^E)_{\pi^{-1}(y),j}$, then the roles move together: $k_{\text{top}}(\rho^\pi) = \pi(k_{\text{top}}(\rho))$ and $\widehat{k}_{\text{best}}^E(\Phi^{E,\pi}) = \pi(\widehat{k}_{\text{best}}^E(\Phi^E))$. Values read *at roles* and symmetric aggregates are therefore unchanged, giving:

**Proposition 4.2** (Permutation invariance of the role-indexed state). *For any* $\pi \in \mathfrak{S}_K$, $\mathrm{RI}\big(\rho^\pi(x), \Phi^{E,\pi}\big) = \mathrm{RI}\big(\rho(x), \Phi^E\big)$. *Equivalently,* $z(x, E)$ *is invariant to coherent relabellings.*

**Corollary 4.1** (Rejector invariance). *For any parameters* $\phi$ *and any* $\pi \in \mathfrak{S}_K$, $g_\perp(x, E^\pi; \phi) = r_\phi\big(\mathrm{RI}(\rho^\pi, \Phi^{E,\pi})\big) = r_\phi\big(\mathrm{RI}(\rho, \Phi^E)\big) = g_\perp(x, E; \phi)$.

*Intuition.* Under *coherent relabelling* (§3.3), the *roles* (model-top, expert-top) move together and the aggregates are symmetric. Hence $z(x, E)$—and thus $g_\perp$—are unchanged. Full proofs in App. E.4.

The following Remark details the multi-expert inference procedure of IFD:

**Remark 4.1** (Multi-expert inference with budget sweep). *Let the available experts be* $\{E_j\}_{j=1}^J$. *Define the model best-class logit* $s(x) := \max_{k \in \mathcal{Y}} g_k(x)$ *and expert deferral logits* $t_j(x) := g_\perp(x, E_j)$. *The expert-specific deferral margins are*

$$\Delta_j(x) := t_j(x) - s(x), \qquad \Delta^\star(x) := \max_{j \in [J]} \Delta_j(x), \qquad j^\star(x) := \arg\max_{j \in [J]} \Delta_j(x).$$

*For any threshold* $\tau \in \mathbb{R}$, *the decision rule is*

$$\hat{y}(x) = \begin{cases} \text{defer to } E_{j^\star(x)}, & \text{if } \Delta^\star(x) \geq \tau, \\ \arg\max_{k \in \mathcal{Y}} g_k(x), & \text{otherwise.} \end{cases} \tag{9}$$

*The default operating point used by standard L2D corresponds to* $\tau = 0$. *Our evaluation varies the deferral budget* $d \in [0, 1]$ *by sweeping* $\tau$: *for a target* $d$, *set* $\tau(d)$ *to the* $(1-d)$-*quantile of* $\{\Delta^\star(x_i)\}_i$ *on the evaluation set, i.e., defer exactly the top* $d$ *fraction by* $\Delta^\star$. *This "winner-takes-all" routing matches the* $K+J$-*logit multi-expert L2D view with* $g_{\perp,j}(x) \equiv t_j(x)$.

## 4.3 UNCERTAINTY-AWARE TRAINING WITH MINIMAL ANNOTATIONS

This subsection describes the context-only training objective that supervises the rejector using the Bayesian per-class profile and formulates deferral supervision as an uncertainty-weighted term.

Here, we reuse the $(K+1)$-way softmax notation $p(\cdot \mid x, \psi)$, $p(\perp \mid x, \psi)$, $Z(x, \psi)$ from §3.2, but instead plug our identity-free state into the embedding by setting $\psi_{\text{IFD}}(x, E) \equiv z(x, E)$.

For a labelled query $(x, y)$ and sampled expert $E$, we avoid query-time expert labels. We supervise deferral only when the expert's peak class matches $y$, and we weight it by a lower-confidence bound $L_y^E = [\mu_y^E - \alpha \sigma_y^E]_+$ ($\alpha \geq 0$):

$$\mathcal{L}_{\text{CE}}^{\text{IFD}}(x, y, E) = -\log p\big(y \mid x, \psi_{\text{IFD}}\big) - L_y^E \mathbb{I}\{\widehat{k}_{\text{best}}^E = y\} \log p\big(\perp \mid x, \psi_{\text{IFD}}\big). \tag{10}$$

This objective is novel in two respects. First, it requires **no query-time expert labels**, as supervision is derived entirely from the context-derived profile $(\widehat{k}_{\text{best}}^E, \mu_y^E, \sigma_y^E)$, which eliminates a significant annotation bottleneck. Second, it incorporates **risk-sensitive weighting** via a lower confidence bound $(L_y^E)$ that reduces supervision from uncertain expert profiles to prevent over-deferral. Prior population-adaptive methods rely on query-time labels and lack this uncertainty-aware mechanism, the effectiveness of which we validate in an ablation study (App. G.2). At training time we aggregate per-expert losses. For a minibatch $\mathcal{B}$ and an optional per-batch subset $\mathcal{E}_\mathcal{B} \subseteq \mathcal{E}$, we optimise

$$\mathcal{L}_{\text{train}} = \frac{1}{|\mathcal{B}|} \sum_{(x,y) \in \mathcal{B}} \frac{1}{|\mathcal{E}_\mathcal{B}|} \sum_{E \in \mathcal{E}_\mathcal{B}} \mathcal{L}_{\text{CE}}^{\text{IFD}}(x, y, E),$$

with $\mathcal{L}_{\text{CE}}^{\text{IFD}}$ from (10). At inference, we score each available $E_j$ and apply the decision rule (9). As $L_y^E$ can alter the proportion of cases deferred, benchmarking against IFD should take $L_y^E = 1$ and evaluate against a sweep of all deferral thresholds.

**Asymptotic analysis and the IFD deferral rule.** We analyse the behaviour of the surrogate loss (10) as the context size grows ($n_y^E \to \infty$). While the induced classifier $h^*(x)$ converges to

the Bayes-optimal predictor, the induced rejector follows a distinct and interpretable rule (proof in App. D.3):

$$r^*_{\text{IFD}}(x, E) = \mathbb{I}\{\max_{y \in \mathcal{Y}} P(Y{=}y \mid x) \leq P(Y{=}k^E_{\text{best}} \mid x)\, \bar{\theta}^E_{k^E_{\text{best}}}\}, \tag{11}$$

where $k^E_{\text{best}}$ is the expert's true best class and $\bar{\theta}^E_{k^E_{\text{best}}}$ their accuracy on it. This differs from the standard 0-1 Bayes rule (Eq. (6)), which compares model confidence to the expert's *expected* accuracy across all classes (i.e., $\sum_y P(Y{=}y \mid x)\,\bar{\theta}^E_y$). The IFD rule instead focuses on the expert's *peak* competence. This is a direct consequence of the loss design, which strategically targets deferral supervision using only context-derived information.

### 4.4 PRIOR KNOWLEDGE AND ROBUSTNESS TO MISSPECIFICATION

The Beta prior used in constructing the Bayesian expert profile (§4.1) provides a way to incorporate prior knowledge about an expert's competence. If an expert supplies a self-assessed per-class accuracy $a^E_y$ with confidence $c^E_y \in [0, 1]$ and a global strength $s \geq 2$,

$$\alpha_y = 1 + c^E_y\, a^E_y\, (s - 2), \qquad \beta_y = 1 + c^E_y\, (1 - a^E_y)\, (s - 2). \tag{12}$$

Low confidence ($c^E_y \simeq 0$) recovers the uniform prior; high confidence increases prior influence. As context accumulates, posteriors override potentially miscalibrated priors:

**Proposition 4.3** (Overcoming prior misspecification). *Let the true accuracy be $\bar{\theta}^E_y$ and the prior mean $\mu_{prior}$ with strength $S_{prior} = \alpha_y + \beta_y$. For any $\epsilon > 0$, $\delta \in (0, 1)$, it suffices that*

$$n^E_y \geq \max\left\{ \frac{S_{prior}\big(|\mu_{prior} - \bar{\theta}^E_y| - \epsilon/2\big)}{\epsilon/2}, \frac{2\ln(2/\delta)}{\epsilon^2} \right\}$$

*to ensure $|\mu^E_y - \bar{\theta}^E_y| < \epsilon$ with probability at least $1 - \delta$.*

Proof is deferred to App. D.1. Practitioners can obtain prior confidence values of experts using structured expert judgment protocols such as Cooke's Classical Model (Colson & Cooke, 2018).

## 5 EXPERIMENTS

We evaluate *Identity-Free Deferral* (IFD) on simulated and real human experts. Our primary research questions (RQs) are:

- **RQ1:** *How does IFD compare to state-of-the-art methods in robustness to* unseen *experts and varying expert diversity?*
- **RQ2:** *How robust is IFD compared to L2D-Pop under* input *distribution shift?*

Additional RQs: **RQ3:** *How does performance scale with increasing context size for IFD versus baselines?*, and **RQ4:** *What is the effect of incorporating prior knowledge under limited context?*, appear in App. G.3 and App. G.4 respectively. We first detail datasets, metrics, expert simulation, and baselines, then present results per RQ. Full implementation details are in App. F.

**Datasets.** We use three medical datasets—`HAM10000` (Abhishek et al., 2025; Tschandl et al., 2018) (dermatology), `Blood Cells` (Acevedo et al., 2020) (microscopy), and `Liver tumours` (Bilic et al., 2023) (radiology)—spanning realistic, high-stakes modalities, plus `ImageNet-16H` (Steyvers et al., 2022) with real human annotations.

**Evaluation under variable deferral budgets.** The default operating point $g_\perp(x, E) \geq \max_k g_k(x)$ hides trade-offs across *deferral budgets*. We therefore use the budgeted multi-expert inference from Remark 4.1: rank by the combined margin $\Delta^\star(x)$, set $\tau(d)$ to the $(1{-}d)$-quantile of $\{\Delta^\star(x_i)\}_i$ on the current test split, and defer those with $\Delta^\star(x) \geq \tau(d)$ (routed to $E_{j^\star(x)}$); otherwise predict with the model. We summarise performance with:

- **AURSAC:** area under the system accuracy (correctness of the composite model–expert system: expert accuracy on deferred cases and model accuracy on non-deferred cases) SysAcc(d) curve.
- **AURDAC:** area under the expert accuracy on deferred cases ExpAcc(d) curve.

$$\mathrm{AURSAC}(d_{\min}, d_{\max}) = \frac{\int_{d_{\min}}^{d_{\max}} \mathrm{SysAcc}(d)\, \mathrm{d}d}{d_{\max} - d_{\min}}, \quad \mathrm{AURDAC}(d_{\min}, d_{\max}) = \frac{\int_{d_{\min}}^{d_{\max}} \mathrm{ExpAcc}(d)\, \mathrm{d}d}{d_{\max} - d_{\min}},$$

for any $[d_{\min}, d_{\max}] \subseteq [0, 1]$, where integrals are approximated numerically.

**Experts.** We evaluate on (i) *real* human annotators (`ImageNet-16H`) and (ii) a simulation that extends Tailor et al. (2024) with instance-dependent accuracy and correlated error types. Instance difficulty is tied to class prototypicality; we consider two archetypes: *Variable Specialists* (accuracy varies sharply with difficulty) and *Stable Specialists* (flatter profiles). As this framework explicitly models instance-level correctness, it theoretically favours query-conditioned population encoders (L2D-Pop), making our comparisons conservative for IFD. Full details are in App. B.

**Baselines.** We compare IFD against:

- **L2D-Pop** (Tailor et al., 2024): SOTA population-adaptive L2D with both query-conditioned (QC) and query-independent (QI) context encoders.
- **Multi-Expert L2D** (Verma et al., 2023): A fixed-expert baseline (ID only; not applicable to unseen experts). This baseline exists to measure the improvement of population-adaptive methods.

## 5.1 RQ1: PERFORMANCE COMPARISON ACROSS EXPERT DIVERSITY

Table 1: Performance of IFD, L2D-Pop (QI, QC variants), and Multi-Expert L2D on healthcare datasets with realistic simulated experts and on ImageNet-16H with real human expert annotations (and scaling image noise). Results are mean±std. AURSAC and AURDAC (abbreviated as SAC and DAC, respectively) across 10 sampled cohorts. ΔSAC reports IFD's improvement over the next best method. Values in **bold** denote column-best performance.

| Dataset | Expert/ Noise Level | Expert Dist. | ΔSAC | IFD *(Ours)* SAC↑ | IFD DAC↑ | L2D-Pop (QC) (Tailor et al. (2024)) SAC↑ | L2D-Pop (QC) DAC↑ | L2D-Pop (QI) (Tailor et al. (2024)) SAC↑ | L2D-Pop (QI) DAC↑ | Multi-Exp. L2D (Verma et al. (2023)) SAC↑ | Multi-Exp. L2D DAC↑ |
|---|---|---|---|---|---|---|---|---|---|---|---|
| `HAM10000` 3/4 ID/OOD Experts $N_C^E = 105$ Cntx Preds Dermatological | Stable | ID | .00 | **.86±.01** | **.88±.03** | .86±.02 | .85±.02 | **.86±.01** | .87±.02 | .85±.00 | .84±.01 |
| | | OOD | +.02 | **.86±.01** | **.86±.04** | .84±.02 | .83±.02 | .84±.01 | .83±.01 | N/A | N/A |
| | Variable | ID | +.02 | **.80±.03** | **.74±.06** | .78±.03 | .70±.08 | .78±.03 | .69±.07 | .74±.01 | .61±.03 |
| | | OOD | +.04 | **.77±.04** | **.69±.09** | .73±.03 | .61±.03 | .73±.03 | .60±.05 | N/A | N/A |
| `Blood Cells` 4/4 ID/OOD Experts $N_C^E = 120$ Cntx Preds Microscopic | Stable | ID | .00 | **.89±.01** | **.84±.03** | .88±.01 | .83±.01 | **.89±.01** | .83±.01 | .87±.01 | .81±.01 |
| | | OOD | +.01 | **.89±.01** | **.85±.03** | .87±.01 | .81±.02 | .88±.00 | .80±.01 | N/A | N/A |
| | Variable | ID | +.03 | **.81±.01** | **.73±.03** | .78±.02 | .62±.03 | .77±.01 | .59±.02 | .75±.01 | .56±.01 |
| | | OOD | +.06 | **.80±.01** | **.69±.03** | .74±.01 | .54±.03 | .74±.01 | .52±.02 | N/A | N/A |
| `Liver tumours` 5/6 ID/OOD Experts $N_C^E = 165$ Cntx Preds Radiological Img. | Stable | ID | +.01 | **.88±.01** | **.81±.02** | .87±.01 | .76±.01 | .86±.01 | .77±.01 | .86±.01 | .76±.02 |
| | | OOD | +.01 | **.88±.01** | **.81±.02** | .87±.01 | .76±.02 | .86±.01 | .77±.01 | N/A | N/A |
| | Variable | ID | +.08 | **.77±.01** | **.63±.02** | .69±.01 | .44±.02 | .69±.01 | .44±.02 | .67±.01 | .43±.01 |
| | | OOD | +.10 | **.78±.01** | **.62±.04** | .68±.01 | .41±.02 | .68±.00 | .42±.01 | N/A | N/A |
| `ImageNet-16H` 2/2 ID/OOD Experts $N_C^E = 240$ Cntx Preds Benchmarking | Noise 95 | ID | +.01 | **.76±.01** | **.92±.01** | .72±.02 | .90±.03 | .75±.01 | .92±.02 | .66±.03 | .90±.02 |
| | | OOD | +.02 | **.76±.01** | **.91±.02** | .73±.02 | .91±.01 | .74±.01 | .89±.01 | N/A | N/A |
| | Noise 110 | ID | +.03 | **.72±.02** | **.85±.02** | .68±.02 | .81±.02 | .69±.01 | .81±.01 | .62±.03 | .80±.03 |
| | | OOD | +.03 | **.71±.02** | **.82±.02** | .67±.02 | .80±.02 | .68±.01 | .80±.03 | N/A | N/A |
| | Noise 125 | ID | +.02 | **.64±.02** | **.69±.04** | .59±.01 | .64±.03 | .62±.01 | .66±.03 | .54±.03 | .63±.02 |
| | | OOD | +.03 | **.66±.02** | **.74±.04** | .61±.02 | .67±.02 | .63±.01 | .68±.02 | N/A | N/A |

We compare IFD and baselines on medical datasets with realistic simulated experts, and on ImageNet-16H with real annotators. Table 1 reports AURSAC$(0, 1)$ and AURDAC$(0, 1)$ (abbreviated as SAC and DAC, respectively) for ID and OOD experts.

**Findings.** IFD consistently matches or exceeds alternatives. For *Stable* experts, all methods are close, with IFD best or tied on SAC/DAC in both ID and OOD. The largest margins appear with *Variable* experts, where diversity and instance-dependence make adaptation harder: IFD improves both overall system accuracy (SAC) and the quality of deferred cases (DAC). Notably, L2D-Pop (QC) and L2D-Pop (QI) exhibit a sharper OOD drop-off. Beyond accuracy, IFD is annotation-efficient. L2D-Pop requires $E \times N$ query annotations, whereas IFD uses only context labels $\sum_E N_C^E$; with $N \gg N_C^E$ (e.g., `Blood Cells`: $N{=}11959$, $N_C^E{=}120$), this yields substantial savings.

## 5.2 RQ2: ROBUSTNESS UNDER INPUT DISTRIBUTION SHIFT

To test robustness to input shift, we use `ImageNet-16H` (Steyvers et al., 2022), which provides human annotations for 16 classes at four degradation levels (80, 95, 110, 125). Annotators are

highly homogeneous (avg. pairwise correlation 0.93), so we apply *Selective Annotation Sampling* (App. C.4) to induce specialisation while preserving human error patterns. We define four groups (Animals, Transport, Household, Tools/Appliances), draw two ID experts from Animals/Household and two OOD experts from Transport/Tools (App. C.5). The backbone is pretrained on 10k ImageNet images disjoint from `ImageNet-16H`. All methods train the system on ID experts at noise 80 and are evaluated on ID/OOD experts across all other levels.

**Findings.** As input images are increasingly corrupted, IFD maintains strong performance and consistently outperforms all baselines for both ID and OOD experts, with gains most pronounced at higher noise (Table 1). This robustness follows from representation design: both L2D-Pop (QC) and L2D-Pop (QI) build latent expert representations by embedding the (potentially distorted) context images, so corruption propagates into the rejector, whereas IFD uses an explicit, query-independent expert profile that avoids re-embedding noisy inputs and yields more stable routing.

## 6  CONCLUSION AND LIMITATIONS

We exposed a structural failure in population-adaptive L2D: *identity-conditioned* shortcuts that violate permutation symmetry and make rejectors brittle to unseen/OOD experts. *Identity-Free Deferral* (IFD) removes identity channels by design—using a query-independent Bayesian per-class profile, a low-dimensional *role-indexed* state, and an uncertainty-aware, context-only loss—yielding architectural permutation invariance, a smaller effective hypothesis class, and an interpretable peak-competence rule. Across medical imaging benchmarks and ImageNet-16H under input shift, IFD matches or exceeds L2D-Pop, with the largest gains on OOD-unseen experts and in scarce-label regimes; it eliminates query-time expert labels and improves monotonically with additional context.

**Limitations.** IFD summarises each expert with a *query-independent, per-class* correctness profile. This brings data efficiency and symmetry, but does not capture *within-class, instance-level* variation in competence, so for highly case-sensitive experts the class-level summary may misestimate the benefit of deferral on particular instances. We accounted for this in our experiments via *instance-dependent expert prediction generation* (and with real annotator variability on ImageNet-16H); despite this stronger stress test, IFD still outperformed query-conditioned baselines, suggesting its explicit profiles support robust routing even without per-instance expert modelling.

We emphasise that our formal invariance results target *coherent relabellings* (one permutation applied consistently to all class-indexed objects). Real OOD shifts can be weaker or more complex (e.g. *expert-only* reindexings, *partial* remappings that affect a subset of classes, or *incoherent* shifts in which expert and model posteriors drift independently). While our proofs do not directly treat these weaker transforms, the architectural diagnosis (identity leakage) together with Cor. 3.1 explains the practical failure mode: any learned dependence on absolute class coordinates makes the rejector sensitive to which coordinate carries a given competence value, so permuting or perturbing coordinates can change the defer/predict decision.

**Future work.** Two complementary paths can reduce this vulnerability: (i) *architectural* remedies that enforce permutation-aware structure (weight-tying, permutation-equivariant encoders, or role-indexed interfaces like IFD) to eliminate absolute-index channels; and (ii) *training* remedies such as within-episode randomised label permutations, targeted partial-remapping augmentations, or regularisers that penalise sensitivity to coordinate swaps. Together these directions aim to turn the architectural certificate of risk into concrete, deployable robustness; a systematic empirical/theoretical study is left to future work.

ACKNOWLEDGEMENTS

We sincerely thank the anonymous reviewers for their thoughtful feedback and constructive criticism of the manuscript. This work was supported by the Engineering and Physical Sciences Research Council. JS and YI are funded by the EPSRC Centre for Doctoral Training in Health Data Science [EP/S02428X/1]. AN, CO and PS acknowledge EPSRC Turing AI Fellowship: Ultra Sound Multi-Modal Video-based Human-Machine Collaboration [EP/X040186/1].

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

CONTENTS OF APPENDIX

# A    EXTENDED LITERATURE REVIEW

**Selective prediction and rejection.** Classical selective prediction (i.e., rejection learning) trains a predictor that may abstain when confidence is low (Chow, 1957; 1970; Cortes et al., 2016). Beyond calibrated abstention (Guo et al., 2017), modern work studies accuracy–coverage trade-offs, and risk-controlling prediction sets (Vovk et al., 2005; Angelopoulos & Bates, 2021). These threads motivate deferral policies that reason about *who* should act.

**Learning to Defer (L2D).** L2D augments selective prediction by comparing the model's prediction to a (costed) expert's decision, optimising system-level loss (Madras et al., 2018; Mozannar & Sontag, 2021). Extensions cover multiple experts (Verma et al., 2023; Mao et al., 2024), triage/budget constraints (Okati et al., 2021), partial or noisy expert signals (Hemmer et al., 2023), top-$k$ setups Montreuil et al. (2025c;b), and two-stage training that treats the classifier distinct modules where the classifier is frozen (Mao et al., 2023; Montreuil et al., 2024). Applications span healthcare (Joshi et al., 2021; Strong et al., 2025), fraud, moderation and safety (Alves et al., 2023; Lykouris & Weng, 2024; Montreuil et al., 2025a).

**Population adaptation and unseen experts.** When the test-time expert is unseen, L2D-Pop meta-learns an expert representation from few-shot *context* and conditions the deferral rejector on it (Tailor et al., 2024). Query-independent encoders summarise the context; query-conditioned variants cross-attend with the current input. Related meta-learning formulations (neural processes and amortised Bayesian adaptation) provide context-to-representation mappings (Kim et al., 2019a; Kim & Hospedales, 2024). Our approach shares the objective—adapting to unseen experts—but differs in representation: instead of learning a holistic latent, we construct an explicit Bayesian, per-class competence profile and expose only *role-indexed, identity-free* statistics to the rejector. This architectural choice enforces permutation symmetry and mitigates identity-conditioned shortcuts.

**Limited expert labels and annotation efficiency.** A practical barrier to L2D is acquiring query-time expert responses. Recent work infers missing labels with EM or latent-variable models (Nguyen et al., 2025), learns expert embeddings to synthesise predictions (Hemmer et al., 2023), or introduces cost-sensitive formulations to manage expert workload (Alves et al., 2024). Our method avoids query-time labels altogether: we train the rejector with a context-only, uncertainty-aware surrogate—using lower-confidence bounds from the expert profile (§4.3)—while retaining the ability to inject priors (§4.4). To our knowledge, this combination of (i) explicit Bayesian expert summaries, (ii) identity-free gating, and (iii) context-only supervision is novel in L2D.

**Human–AI complements beyond deferral.** Another line *complements* rather than defers to humans—e.g., routing to humans with decision support, or combining predictions through learned aggregation, allocation, or explanation mechanisms (Zhang et al., 2025). These works differ in objective (joint performance, trust, or workload) and interface (advice vs. hand-off); our focus is explicit hand-off under class-aware competence.

**Agnostic and permutation-aware learning.** IFD echoes agnostic/meta-learning that targets robustness to task/domain variation (e.g., MAML, TAML, and agnostic federated learning) (Finn et al., 2017; Jamal et al., 2018; Mohri et al., 2019). Technically, our contribution is architectural: we enforce invariance to coherent class relabellings by operating on role-indexed statistics and symmetric rejectors. This mirrors recent interest in symmetry, invariance and equivariance as inductive biases for OOD generalisation (Bloem-Reddy & Teh, 2020; Bronstein et al., 2021). We show that standard L2D-Pop parameterisations are *generically* non-invariant.

**Positioning.** Compared to L2D-Pop, our explicit profile: (i) reduces variance at small context sizes, (ii) removes identity channels by construction, (iii) naturally incorporates informative priors, and (iv) scales with additional context at test time. Compared to limited-label L2D variants, we eliminate query-time labels while preserving adaptation to unseen experts. Compared to complement-style collaboration, we target the Bayes comparison between the model's confidence and the expert's competence and implement it with identity-free features.

# B    INSTANCE-DEPENDENT EXPERT SIMULATION PROTOCOL

To enhance the fidelity of our evaluation and address the limitations of expert simulations that assume uniform intra-class accuracy, we developed a more realistic, instance-dependent expert simu-

lation protocol than that utilised in Tailor et al. (2024). This protocol models an expert's accuracy as a function of instance difficulty, allowing for a more granular and challenging assessment of deferral strategies.

## B.1 MOTIVATION FOR A MORE REALISTIC SIMULATION

Learning to Defer (L2D) systems are most valuable when they can adapt to the nuanced capabilities of human experts. Prior work, including the baseline simulation protocol from Tailor et al. (2024), often models expert performance at the class level. For instance, an expert might be assigned a fixed accuracy of 95% on their specialty class and a lower accuracy on all other classes.

This abstraction, while useful for studying generalisation to OOD expert specialisations, overlooks a critical aspect of real-world expertise: performance might not be uniform within a class. An expert radiologist, for example, will find it trivial to identify a classic, textbook example of a tumour (a "prototypical" case) but may struggle with an atypical presentation of the same disease.

By assuming fixed per-class accuracy, simpler simulations cannot test a deferral system's ability to distinguish between easy and hard instances for an expert. To address this, we introduce a simulation framework where an expert's accuracy for any given sample is a direct function of that sample's *prototypicality* within its true class. This allows us to model experts with more complex and realistic behaviours.

**A more challenging testbed for our method.** It is important to note that this instance-dependent simulation protocol creates a more challenging evaluation landscape specifically for our proposed method, IFD, while theoretically favouring the baseline, L2D-Pop. The L2D-Pop framework is designed to learn a holistic, latent representation of an expert that implicitly models the complex function mapping an instance to a prediction ($P(M|Y, X)$). Our new simulation generates data that explicitly follows such an instance-dependent function. In contrast, IFD makes a principled abstraction, modelling performance at the class level ($P(M|Y)$) for robustness and generalisability. Therefore, by evaluating IFD in a setting where the ground-truth expert behaviour is instance-dependent, we are stress-testing our method against a baseline whose core assumptions are better aligned with the simulation's design. Strong performance under these conditions would provide compelling evidence for the robustness of our approach.

## B.2 METHODOLOGY

Our simulation methodology consists of three main stages: (1) Quantifying instance prototypicality using a learned feature space, (2) modelling instance-dependent expert accuracy using this prototypicality score, and (3) Generating stochastic, correlated expert predictions based on the modelled accuracy.

### B.2.1 QUANTIFYING INSTANCE PROTOTYPICALITY

We define the prototypicality of an instance as its proximity to the central tendency of its class in a learned feature space. To achieve this, we first train a standard classifier on the task dataset to serve as a feature extractor.

**1. Feature Extraction:** Let $\mathcal{F}$ be the feature extractor (i.e., the penultimate layer of a trained classifier). For each image $\mathbf{x}_i$ in the training dataset $\mathcal{D}_{train}$, we extract its feature vector $\mathbf{z}_i = \mathcal{F}(\mathbf{x}_i)$.

**2. Class Centroid Calculation:** For each class $k \in \{1, \ldots, K\}$, we compute the mean feature vector, or centroid $\mathbf{c}_k$, from all training samples belonging to that class:

$$\mathbf{c}_k = \frac{1}{|\mathcal{D}_{train,k}|} \sum_{\mathbf{x}_i \in \mathcal{D}_{train,k}} \mathcal{F}(\mathbf{x}_i) \tag{13}$$

where $\mathcal{D}_{train,k}$ is the subset of training data with label $k$.

**3. Prototypicality Score:** The prototypicality $p(\mathbf{x}_i)$ of an instance $\mathbf{x}_i$ with true label $y_i = k$ is defined by its normalised distance to its class centroid. First, we find the maximum distance $d_{max,k}$ between any training instance of class $k$ and its centroid:

$$d_{max,k} = \max_{\mathbf{x}_j \in \mathcal{D}_{train,k}} ||\mathcal{F}(\mathbf{x}_j) - \mathbf{c}_k||_2 \tag{14}$$

The prototypicality score $p(\mathbf{x}_i) \in [0, 1]$ is then calculated as:

$$p(\mathbf{x}_i) = 1 - \min\left(1, \frac{||\mathcal{F}(\mathbf{x}_i) - \mathbf{c}_k||_2}{d_{max,k}}\right) \tag{15}$$

A score of $p(\mathbf{x}_i) \approx 1$ indicates a highly prototypical (easy) instance, while $p(\mathbf{x}_i) \approx 0$ indicates a highly atypical (hard) instance.

### B.2.2 INSTANCE-DEPENDENT ACCURACY MODELLING

We model an expert's instance-specific accuracy using a sigmoid function, which takes the prototypicality score as input. This allows us to define expert behaviour with two intuitive parameters. Let $A(\mathbf{x}_i|E)$ be the accuracy of expert $E$ on instance $\mathbf{x}_i$. We model this as:

$$A(\mathbf{x}_i|E) = \sigma(\alpha \cdot p(\mathbf{x}_i) + \beta) = \frac{1}{1 + e^{-(\alpha \cdot p(\mathbf{x}_i) + \beta)}} \tag{16}$$

where:

- $\alpha$ controls the steepness of the curve. A high $\alpha$ models an expert whose performance is highly sensitive to instance difficulty (i.e., a large gap between performance on easy vs. hard cases). A low $\alpha$ models an expert with more consistent performance.

- $\beta$ controls the horizontal shift of the curve, effectively setting the baseline accuracy for the most atypical cases ($p = 0$).

To make these parameters more intuitive, we derive them from two target accuracies: $\mathrm{acc}_{\text{easy}}$ (the desired accuracy for the most prototypical cases, $p = 1$) and $\mathrm{acc}_{\text{hard}}$ (the desired accuracy for the most atypical cases, $p = 0$). Using the inverse sigmoid (logit) function, we solve for $\alpha$ and $\beta$:

$$\beta = \mathrm{logit}(\mathrm{acc}_{\text{hard}}) = -\log\left(\frac{1}{\mathrm{acc}_{\text{hard}}} - 1\right) \tag{17}$$

$$\alpha = \mathrm{logit}(\mathrm{acc}_{\text{easy}}) - \beta \tag{18}$$

### B.2.3 STOCHASTIC AND CORRELATED PREDICTION GENERATION

The final simulation process for an instance $\mathbf{x}_i$ with true label $y_i$, evaluated against a population of experts, is as follows:

1. Compute the instance's prototypicality score $p(\mathbf{x}_i)$.
2. Generate a single instance-specific random number $u_i \sim U(0, 1)$. This value represents a latent solvability factor for the instance and will be used for all experts.
3. For each expert $E$ in the population:
   (a) Determine if the instance belongs to the expert's specialty class ($y_i = k_E$).
   (b) Select the appropriate parameters: $(\alpha_{spec}, \beta_{spec})$ if $y_i = k_E$, or $(\alpha_{non-spec}, \beta_{non-spec})$ otherwise.
   (c) Calculate the expert's instance-specific accuracy $A(\mathbf{x}_i|E)$ using the selected parameters and equation (4).
   (d) If $u_i < A(\mathbf{x}_i|E)$, the expert's prediction is correct ($m_i = y_i$).
   (e) Otherwise, the expert's prediction is incorrect. The specific incorrect class, $m_i$, is then drawn from a shared, instance-specific error distribution.

**Correlating the Type of Error.** A simple simulation might draw an incorrect prediction uniformly from the set of wrong answers. To create more realistic error patterns, where experts tend to agree on the same incorrect class for a confusing instance, we model the error distribution. When an expert fails to predict correctly (i.e., $u_i \geq A(\mathbf{x}_i|E)$), we make the probability of predicting an incorrect class $j$ inversely proportional to the feature space distance to that class's centroid. This ensures that an instance is most likely to be confused with a visually or semantically similar class. The incorrect prediction $m_i$ is drawn from the distribution:

$$P(m_i = j|\text{incorrect}, \mathbf{x}_i) = \mathrm{softmax}(-||\mathcal{F}(\mathbf{x}_i) - \mathbf{c}_j||_2) = \frac{e^{-||\mathcal{F}(\mathbf{x}_i) - \mathbf{c}_j||_2}}{\sum_{l \in \mathcal{Y} \setminus \{y_i\}} e^{-||\mathcal{F}(\mathbf{x}_i) - \mathbf{c}_l||_2}} \tag{19}$$

for all incorrect classes $j \in \mathcal{Y} \setminus \{y_i\}$. Since this distribution depends only on the instance $\mathbf{x}_i$ and the class centroids, all experts who make an error on this instance will sample their incorrect prediction from the same distribution, thus correlating their specific mistakes.

**Reproducibility.** The instance-specific variable $u_i$ is generated deterministically as a function of the instance index and a global seed, so that all experts evaluating the same instance share the same correctness draw while results remain reproducible.

**Sampling from the shared error distribution.** When $u_i \geq A(\mathbf{x}_i|E)$, the incorrect class is sampled independently for each expert from the same, instance-specific distribution in Eq. (7), with the true class masked (probability zero). This induces correlated error *types* across experts without forcing identical wrong labels. If one wishes to enforce identical wrong labels, a single shared draw from the distribution can be used for all experts on that instance.

### B.3 EXPERT ARCHETYPES FOR EVALUATION

To rigorously test the capabilities of a deferral system, we simulate populations composed of distinct expert archetypes, as summarised in Table 2. These archetypes are defined by their performance on their specialty class versus non-specialty classes, and critically, how sensitive their performance is to instance difficulty (i.e., the gap between $\text{acc}_{\text{easy}}$ and $\text{acc}_{\text{hard}}$).

Table 2: Parameters defining the expert archetypes used in the simulations. Archetypes are defined by the target accuracy on the easiest ($\text{acc}_{\text{easy}}$, $p = 1$) and hardest ($\text{acc}_{\text{hard}}$, $p = 0$) instances, which determine the sigmoid parameters $\alpha$ and $\beta$ via Equations (5) and (6).

| Archetype | Context | $\text{acc}_{\text{easy}}$ | $\text{acc}_{\text{hard}}$ | $\alpha$ | $\beta$ |
|---|---|---|---|---|---|
| Variable Specialist | Specialty | 0.99 | 0.50 | 4.595 | 0.000 |
| | Non-Specialty | 0.80 | 0.20 | 2.773 | -1.386 |
| Stable Specialist | Specialty | 0.99 | 0.85 | 2.860 | 1.735 |
| | Non-Specialty | 0.85 | 0.70 | 0.886 | 0.847 |

The **Variable Specialist** represents an expert whose accuracy drops significantly when faced with atypical examples. In contrast, the **Stable Specialist** maintains high accuracy even on difficult instances, representing a more consistently reliable expert.

## C EVALUATION USING REAL-WORLD HUMAN ANNOTATIONS

A critical requirement for validating L2D systems, particularly those intended for high-stakes domains, is the assessment of performance using real-world human annotations. While simulated experts (Appendix B) allow for controlled stress-testing, benchmarking against actual human behaviour ensures robustness to the nuances inherent in human decision-making. We utilise the ImageNet-16H (Steyvers et al., 2022) dataset, which provides multiple human annotations per image.

### C.1 MOTIVATION AND CHALLENGES

The core contribution of IFD is its robust generalisation to OOD experts–those whose specialisation profiles differ significantly from experts encountered during training. Evaluating this capability necessitates datasets containing diverse, specialised expert behaviours. However, standard multi-annotator datasets, often collected via crowd-sourcing for general recognition tasks, frequently exhibit high homogeneity in annotator behaviour. This homogeneity precludes their direct use in evaluating OOD specialisation generalisation, as there is insufficient diversity to define distinct ID and OOD specialisation profiles.

To address this limitation while adhering to the requirement for real-world data, we employ a semi-synthetic evaluation strategy. This strategy introduces the necessary specialisation structure for OOD testing while strictly utilising the real, instance-dependent error patterns present in the datasets.

C.2    ANALYSIS AND QUANTIFICATION OF ANNOTATOR HOMOGENEITY

We first empirically assessed the diversity of specialisation within the raw datasets to justify the semi-synthetic approach.

**Methodology.** We define the "behavioural fingerprint" of an annotator as their normalised confusion matrix. This matrix captures $P$(Chosen Label|True Label), representing both their classwise accuracy and their specific patterns of mistakes. The analysis proceeds as follows:

1. **Fingerprint Calculation:** The confusion matrix for each individual annotator is calculated and normalised by the true class (rows).

2. **Vectorisation:** The $K \times K$ normalised matrix is flattened into a $K^2$ vector.

3. **Homogeneity Quantification:** We calculate the average pairwise correlation between all annotator vectors.

4. **Characterisation:** K-Means clustering is applied to the vectors to identify distinct behavioural subgroups within the population.

**Findings.** The analysis confirmed extreme homogeneity in specialisation profiles, with an average pairwise correlation of 0.926 for ImageNet-16H. Clustering analysis revealed that the primary axis of variation among annotators was overall skill level (e.g., separating high-accuracy annotators from low-accuracy or noisy annotators), rather than distinct areas of specialisation.

**Implications.** The high homogeneity empirically demonstrates that these datasets cannot be used "as-is" to evaluate generalisation across diverse specialisation profiles, validating the necessity of the semi-synthetic approach detailed below.

C.3    PREPARATION OF THE ANNOTATION POOL (FILTERING)

To ensure that synthesised expert behaviours are representative of competent experts rather than noise or inattention, the raw annotation pool was filtered prior to synthesis. Annotators identified in the clustering analysis (Section C.2) as belonging to low-skill clusters (e.g., those performing near random chance) were excluded. This filtering step ensures that when a synthesised expert makes a mistake, the error is drawn from a plausible mistake made by a competent human annotator, maintaining the realism of the evaluation.

C.4    SEMI-SYNTHETIC EXPERT GENERATION: SELECTIVE ANNOTATION SAMPLING

We employ a methodology termed **Selective Annotation Sampling** to synthesise specialised experts. The defining principle of this approach is that it **does not introduce artificial errors**. 100% of the annotations used in the evaluation are actual human responses provided for the specific image being considered.

**Algorithm Description.** The synthesis process involves the following steps:

1. **Define Target Profiles:** We define the desired specialisation profiles (e.g., ID and OOD groups) and the target accuracies for specialty and non-specialty classes (e.g., $A_{high}$ and $A_{low}$).

2. **Organise the Annotation Pool:** For every image $X_i$ in the dataset, the available human annotations (from the filtered pool) are partitioned into two sets:
   - `Correct_Pool`: Annotations matching the true label $y_i$.
   - `Incorrect_Pool`: Annotations not matching $y_i$.

3. **Synthesise Expert Annotations:** For a synthesised expert $E_{synth}$ evaluating image $X_i$:
   (a) Determine the target accuracy $A_{target}$ based on the true label $y_i$ and the expert's defined profile.
   (b) Stochastically determine the desired outcome:  `Should_Be_Correct` = Bernoulli($A_{target}$).
   (c) **Selective Sampling with Constraints:**

- If the desired outcome is Correct: Attempt to sample from `Correct_Pool`. If `Correct_Pool` is empty (a difficult image where all available humans erred), the expert must sample from `Incorrect_Pool`.
- If the desired outcome is Incorrect: Attempt to sample from `Incorrect_Pool`. If `Incorrect_Pool` is empty (an easy image where all available humans were correct), the expert must sample from `Correct_Pool`.

This methodology generates experts with distinct specialisation profiles while ensuring that all errors are authentic human mistakes, thereby preserving the crucial instance-dependent error patterns of the original data.

## C.5 Experimental Designs for OOD generalisation

Using Selective Annotation Sampling, we construct several experiments to test OOD generalisation on ImageNet-16H.

### ImageNet-16H Design: OOD generalisation under Distribution Shift

ImageNet-16H contains 16 classes evaluated across four distinct levels of Gaussian noise added to the images, indexed as 80 (lowest noise), 95, 110, and 125 (highest noise). This structure allows us to evaluate OOD generalisation concurrently with robustness to input distribution shift. We utilised $A_{high} = 0.99$ and $A_{low} = 0.40$.

**Expert Group Definitions.** We partitioned the 16 classes into four specialisation groups (indices correspond to the experimental mapping):

- **Group 1 (Animals):** Indices [1, 3, 4, 10, 14] (dog, bird, bear, elephant, cat).
- **Group 2 (Transport):** Indices [5, 8, 9, 12, 13] (airplane, car, boat, truck, bicycle).
- **Group 3 (Household):** Indices [0, 6, 7, 11] (chair, clock, bottle, knife).
- **Group 4 (Tools/Appliances):** Indices [2, 6, 15] (oven, keyboard, knife).

Note the overlap of 'knife' (Index 6) between Group 3 and Group 4, reflecting potential overlaps in real-world expertise.

**ID/OOD Split and Training Protocol.** We utilised a multi-group ID/OOD split:

- **ID Cohort (Training):** Comprised of 4 synthesised experts: 2 specialised in Group 1 (Animals) and 2 specialised in Group 3 (Household).
- **OOD Cohort (Testing):** Comprised of 4 synthesised experts: 2 specialised in Group 2 (Transport) and 2 specialised in Group 4 (Tools/Appliances).

To evaluate robustness to input distribution shift:

1. **Classifier Pretraining:** The classifier backbone was pretrained on 10,000 standard ImageNet images disjoint from the ImageNet-16H dataset.
2. **L2D Training:** The deferral policies were trained using annotations from the ID Cohort, exclusively on images with the lowest noise level (80).
3. **Evaluation (Distribution Shift):** The trained systems were evaluated on both the ID and OOD cohorts across all three noise levels (95, 110, 125) without retraining or fine-tuning on the higher noise levels.

## C.6 Verification of Synthesis and the Constraint Effect

To verify the synthesis process, we calculate the average realised confusion matrix for each synthesised cohort and compare the realised classwise accuracy to the target accuracy.

**The Constraint Effect.** During verification, it is frequently observed that the Realised Accuracy is higher than the Target Accuracy, particularly when the target is low (e.g., realised 0.75 when the target was 0.60). This phenomenon, termed the "Constraint Effect," arises directly from the

constraint that only existing human annotations can be utilised (Step 3c in Section C.4). If the algorithm aims for an incorrect answer on an image where all filtered human annotators were correct (`Incorrect_Pool` is empty), it is forced to select the correct answer. This effect confirms the fidelity of the simulation to the real data, as it accurately reflects the underlying difficulty of the images for the human pool and avoids introducing artificial errors on trivially easy instances.

# D  GENERAL PROOFS

**Theorem D.1** (Hoeffding's Inequality). *Let $Z_1, \ldots, Z_n$ be random independent, identically distributed random variables, such that $0 \leq Z_i \leq 1$. Then,*

$$\Pr\left[\left|\frac{1}{n}\sum_{i=1}^{n} Z_i - \mathbb{E}[Z]\right| > \epsilon\right] \leq \delta = 2\exp(-2n\epsilon^2)$$

**Theorem D.2** (Boole's Inequality). *For any countable collection of events $A_1, A_2, \ldots, A_n$ in a probability space,*

$$\Pr\left(\bigcup_{i=1}^{n} A_i\right) \leq \sum_{i=1}^{n} P(A_i).$$

## D.1  PROOF OF PROPOSITION 4.1

**Proposition 4.1** (Peak-class identification). *Let an arbitrary expert have true but unknown per-class accuracies $\bar{\theta}_k \in [0,1]$ (we drop the E notation here). Let $k_{best} = \arg\max_k \bar{\theta}_k$ be the unique class of maximal expertise. As the number of context samples $n_k \to \infty$ for each class $k$, the posterior mean estimates $\mu_k$ converge almost surely to $\bar{\theta}_k$, and the selected expertise class $\arg\max_k \mu_k$ converges almost surely to $k_{best}$. Moreover, if each class $k$ is observed at least $n$ times, where*

$$n \geq \frac{\ln\left(\frac{2K}{\delta}\right)}{2\left(\frac{\Delta_{acc}}{2}\right)^2}, \quad \text{with } \Delta_{acc} := \bar{\theta}_{k_{best}} - \max_{k \neq k_{best}} \bar{\theta}_k.$$

*then with probability at least $1 - \delta$, the posterior means satisfy $\mu_{k_{best}} > \mu_k$ for all $k \neq k_{best}$; that is, the correct expertise class is identified.*

*Proof.* **Step 1: Posterior convergence**  For each class $k$, the expert's true accuracy $\bar{\theta}_k \in [0,1]$ is modelled with a Beta prior:
$$\theta_k \sim \text{Beta}(\alpha_k, \beta_k).$$
After observing $n_k$ samples for class $k$, of which $t_k$ are correct, the posterior becomes:
$$\theta_k \mid \mathcal{D}_C \sim \text{Beta}(\alpha_k + t_k, \beta_k + n_k - t_k),$$
with posterior mean:
$$\mu_k = \frac{\alpha_k + t_k}{\alpha_k + \beta_k + n_k}.$$
By the Law of Large Numbers, the empirical accuracy $\hat{\theta}_k := t_k/n_k$ converges almost surely to the true accuracy $\bar{\theta}_k$ as $n_k \to \infty$. Since the prior's influence vanishes in the limit, the posterior mean $\mu_k$ also converges almost surely to $\bar{\theta}_k$.

**Step 2: Consistent identification of the expert class**  Let $k_{\text{best}} = \arg\max_k \bar{\theta}_k$ be the unique best class. Since $\mu_k \xrightarrow{a.s.} \bar{\theta}_k$, we have: $\arg\max_k \mu_k \xrightarrow{a.s.} k_{\text{best}}$. That is, the class selected by maximizing the posterior mean converges almost surely to the correct class.

**Step 3: Finite-sample guarantee using Hoeffding's inequality**  We apply Hoeffding's inequality (Thm. D.1):
$$P\left(\left|\hat{\theta}_k - \bar{\theta}_k\right| \geq \epsilon\right) \leq 2\exp(-2\epsilon^2 n_k).$$
To ensure correct identification, we require:
$$\left|\hat{\theta}_k - \bar{\theta}_k\right| < \frac{\Delta_{\text{acc}}}{2}, \quad \forall k,$$

where $\Delta_{\mathsf{acc}} = \bar{\theta}_{k_{\mathsf{best}}} - \max_{k \neq k_{\mathsf{best}}} \bar{\theta}_k$. This guarantees:

$$\hat{\theta}_{k_{\mathsf{best}}} > \hat{\theta}_k, \quad \forall k \neq k_{\mathsf{best}}.$$

Setting $\epsilon = \Delta_{\mathsf{acc}}/2$ and assuming $n_k \geq n$ for all $k$, Hoeffding gives:

$$P\left(\exists k \text{ s.t. } \left|\hat{\theta}_k - \bar{\theta}_k\right| \geq \frac{\Delta_{\mathsf{acc}}}{2}\right) \leq 2K \exp\left(-2\left(\frac{\Delta_{\mathsf{acc}}}{2}\right)^2 n\right).$$

We want this probability $\leq \delta$, so solve:

$$2K \exp\left(-2\left(\frac{\Delta_{\mathsf{acc}}}{2}\right)^2 n\right) \leq \delta \quad \Rightarrow \quad n \geq \frac{\ln\left(\frac{2K}{\delta}\right)}{2\left(\frac{\Delta_{\mathsf{acc}}}{2}\right)^2}.$$

Thus, with this many samples per class, we have $\mu_{k_{\mathsf{best}}} > \mu_k$ for all $k \neq k_{\mathsf{best}}$ with probability $\geq 1 - \delta$. $\qquad\square$

## D.2 PROOF OF PROPOSITION 4.3

**Lemma D.1** (Recovery Rate from Misspecified Priors). *Let $\bar{\theta}_k^E$ be the true accuracy of expert $E$ for class $k$, and suppose a potentially misspecified prior $Beta(\alpha_k^{prior}, \beta_k^{prior})$ is used. After observing $n_y^E$ context samples for class $k$ resulting in $t_k^E$ correct predictions, the absolute error of the posterior mean estimate $\mu_y^E$ is bounded by:*

$$|\mu_y^E - \bar{\theta}_k^E| \leq \frac{S_{prior}}{S_{prior} + n_y^E}|\mu_{prior} - \bar{\theta}_k^E| + \frac{n_y^E}{S_{prior} + n_y^E}|\hat{\theta}_k^{(E)} - \bar{\theta}_k^E|$$

*where $S_{prior} = \alpha_k^{prior} + \beta_k^{prior}$, $\mu_{prior} = \frac{\alpha_k^{prior}}{S_{prior}}$, and $\hat{\theta}_k^{(E)} = \frac{t_k^E}{n_y^E}$. Furthermore, as $n_y^E \to \infty$, the posterior mean converges to the true accuracy: $\mu_y^E \to \bar{\theta}_k^E$.*

*Proof.* The posterior mean $\mu_y^E$ after observing $t_k^E$ successes in $n_y^E$ trials, given the prior $Beta(\alpha_k^{prior}, \beta_k^{prior})$, is:

$$\mu_y^E = \frac{\alpha_k^{prior} + t_k^E}{\alpha_k^{prior} + \beta_k^{prior} + n_y^E}$$

Using the definitions $S_{\mathsf{prior}} = \alpha_k^{prior} + \beta_k^{prior}$ and $\hat{\theta}_k^{(E)} = t_k^E/n_y^E$, we can rewrite the posterior mean:

$$\mu_y^E = \frac{\alpha_k^{prior} + n_y^E \hat{\theta}_k^{(E)}}{S_{\mathsf{prior}} + n_y^E}$$

$$= \frac{\alpha_k^{prior}}{S_{\mathsf{prior}} + n_y^E} + \frac{n_y^E \hat{\theta}_k^{(E)}}{S_{\mathsf{prior}} + n_y^E}$$

Recalling $\mu_{\mathsf{prior}} = \alpha_k^{prior}/S_{\mathsf{prior}}$, we have $\alpha_k^{prior} = S_{\mathsf{prior}}\mu_{\mathsf{prior}}$. Substituting this gives:

$$\mu_y^E = \frac{S_{\mathsf{prior}}\mu_{\mathsf{prior}}}{S_{\mathsf{prior}} + n_y^E} + \frac{n_y^E \hat{\theta}_k^{(E)}}{S_{\mathsf{prior}} + n_y^E}$$

$$= \frac{S_{\mathsf{prior}}}{S_{\mathsf{prior}} + n_y^E}\mu_{\mathsf{prior}} + \frac{n_y^E \hat{\theta}_k^{(E)}}{S_{\mathsf{prior}} + n_y^E}$$

Let $w_{\mathsf{prior}} = \frac{S_{\mathsf{prior}}}{S_{\mathsf{prior}} + n_y^E}$ and $w_{\mathsf{data}} = \frac{n_y^E}{S_{\mathsf{prior}} + n_y^E}$. Note that $w_{\mathsf{prior}} + w_{\mathsf{data}} = 1$. Thus, the posterior mean is a weighted average of the prior mean and the empirical accuracy:

$$\mu_y^E = w_{\mathsf{prior}}\mu_{\mathsf{prior}} + w_{\mathsf{data}}\hat{\theta}_k^{(E)}$$

Now consider the absolute error $|\mu_y^E - \bar{\theta}_k^E|$. Since $w_{\text{prior}} + w_{\text{data}} = 1$, we can write $\bar{\theta}_k^E = (w_{\text{prior}} + w_{\text{data}})\bar{\theta}_k^E$.

$$|\mu_y^E - \bar{\theta}_k^E| = |(w_{\text{prior}}\mu_{\text{prior}} + w_{\text{data}}\hat{\theta}_k^{(E)}) - (w_{\text{prior}} + w_{\text{data}})\bar{\theta}_k^E|$$

$$= |w_{\text{prior}}(\mu_{\text{prior}} - \bar{\theta}_k^E) + w_{\text{data}}(\hat{\theta}_k^{(E)} - \bar{\theta}_k^E)|$$

Applying the triangle inequality ($|a + b| \leq |a| + |b|$):

$$|\mu_y^E - \bar{\theta}_k^E| \leq |w_{\text{prior}}(\mu_{\text{prior}} - \bar{\theta}_k^E)| + |w_{\text{data}}(\hat{\theta}_k^{(E)} - \bar{\theta}_k^E)|$$

$$= w_{\text{prior}}|\mu_{\text{prior}} - \bar{\theta}_k^E| + w_{\text{data}}|\hat{\theta}_k^{(E)} - \bar{\theta}_k^E| \quad (\text{since } w_{\text{prior}}, w_{\text{data}} \geq 0)$$

Substituting back the definitions of $w_{\text{prior}}$ and $w_{\text{data}}$ yields the error bound:

$$|\mu_y^E - \bar{\theta}_k^E| \leq \frac{S_{\text{prior}}}{S_{\text{prior}} + n_y^E}|\mu_{\text{prior}} - \bar{\theta}_k^E| + \frac{n_y^E}{S_{\text{prior}} + n_y^E}|\hat{\theta}_k^{(E)} - \bar{\theta}_k^E|$$

This proves the first part of the proposition (i.e., bounds of the absolute error of the posterior mean estimate).

For the convergence analysis, consider the limit as $n_y^E \to \infty$:

$$\lim_{n_y^E \to \infty} w_{\text{prior}} = \lim_{n_y^E \to \infty} \frac{S_{\text{prior}}}{S_{\text{prior}} + n_y^E} = 0$$

$$\lim_{n_y^E \to \infty} w_{\text{data}} = \lim_{n_y^E \to \infty} \frac{n_y^E}{S_{\text{prior}} + n_y^E} = 1$$

By the Law of Large Numbers (specifically, the Strong Law for Bernoulli trials), the empirical accuracy converges almost surely to the true accuracy:

$$\hat{\theta}_k^{(E)} = \frac{t_k^E}{n_y^E} \xrightarrow{\text{a.s.}} \bar{\theta}_k^E \quad \text{as } n_y^E \to \infty$$

Therefore, taking the limit of the posterior mean:

$$\lim_{n_y^E \to \infty} \mu_y^E = \lim_{n_y^E \to \infty} \left( w_{\text{prior}}\mu_{\text{prior}} + w_{\text{data}}\hat{\theta}_k^{(E)} \right)$$

$$= \left( \lim_{n_y^E \to \infty} w_{\text{prior}} \right) \mu_{\text{prior}} + \left( \lim_{n_y^E \to \infty} w_{\text{data}} \right) \left( \lim_{n_y^E \to \infty} \hat{\theta}_k^{(E)} \right)$$

$$= (0) \cdot \mu_{\text{prior}} + (1) \cdot \bar{\theta}_k^E$$

$$= \bar{\theta}_k^E$$

Thus, the posterior mean converges to the true accuracy regardless of the initial prior specification.
$\square$

**Proposition 4.3** (Overcoming prior misspecification). *Let $\bar{\theta}_k^E$ be the true accuracy of expert $E$ for class $k$, with prior $Beta(\alpha_k^{prior}, \beta_k^{prior})$ having mean $\mu_{prior}$ and strength $S_{prior} = \alpha_k^{prior} + \beta_k^{prior}$. For any $\epsilon > 0$ and $\delta \in (0, 1)$, the number of context samples $n_y^E$ needed to ensure the posterior mean $\mu_y^E$ satisfies $|\mu_y^E - \bar{\theta}_k^E| < \epsilon$ with probability at least $1 - \delta$ is bounded by:*

$$n_y^E \geq \max \left\{ \frac{S_{prior}(|\mu_{prior} - \bar{\theta}_k^E| - \epsilon/2)}{\epsilon/2}, \frac{2\ln(2/\delta)}{\epsilon^2} \right\}$$

*The first term accounts for overcoming prior bias (relevant if $|\mu_{prior} - \bar{\theta}_k^E| > \epsilon/2$) and the second accounts for statistical uncertainty via Hoeffding's inequality.*

*Proof.* From Lemma D.1, we have the bound:

$$|\mu_y^E - \bar{\theta}_k^E| \leq \underbrace{\frac{S_{\text{prior}}}{S_{\text{prior}} + n_y^E}|\mu_{\text{prior}} - \bar{\theta}_k^E|}_{\text{Term 1: Prior Bias}} + \underbrace{\frac{n_y^E}{S_{\text{prior}} + n_y^E}|\hat{\theta}_k^{(E)} - \bar{\theta}_k^E|}_{\text{Term 2: Data Estimation Error}}$$

Our objective is to find $n_y^E$ such that $|\mu_y^E - \bar{\theta}_k^E| < \epsilon$ with probability at least $1 - \delta$. A sufficient condition is to ensure that each term on the right-hand side is bounded by $\epsilon/2$.

**Bounding Term 1 (Prior Bias):** We require:

$$\frac{S_{\text{prior}}|\mu_{\text{prior}} - \bar{\theta}_k^E|}{S_{\text{prior}} + n_y^E} < \frac{\epsilon}{2}$$

If $|\mu_{\text{prior}} - \bar{\theta}_k^E| \leq \epsilon/2$, the condition is trivially satisfied for the term's contribution for large enough $n_y^E$ (as the fraction $\frac{S_{\text{prior}}}{S_{\text{prior}} + n_y^E} \to 0$). Assume $|\mu_{\text{prior}} - \bar{\theta}_k^E| > \epsilon/2$. Multiplying both sides by $(S_{\text{prior}} + n_y^E)$ (which is positive):

$$S_{\text{prior}}|\mu_{\text{prior}} - \bar{\theta}_k^E| < \frac{\epsilon}{2}(S_{\text{prior}} + n_y^E)$$

Rearranging to solve for $n_y^E$:

$$S_{\text{prior}}|\mu_{\text{prior}} - \bar{\theta}_k^E| < \frac{\epsilon}{2}S_{\text{prior}} + \frac{\epsilon}{2}n_y^E$$

$$S_{\text{prior}}|\mu_{\text{prior}} - \bar{\theta}_k^E| - \frac{\epsilon}{2}S_{\text{prior}} < \frac{\epsilon}{2}n_y^E$$

$$S_{\text{prior}}(|\mu_{\text{prior}} - \bar{\theta}_k^E| - \epsilon/2) < \frac{\epsilon}{2}n_y^E$$

Since we assumed $|\mu_{\text{prior}} - \bar{\theta}_k^E| > \epsilon/2$, the term in brackets is positive. Dividing by $\epsilon/2$:

$$n_y^E > \frac{S_{\text{prior}}(|\mu_{\text{prior}} - \bar{\theta}_k^E| - \epsilon/2)}{\epsilon/2}$$

Let $N_{\text{prior}}(\epsilon) = \frac{S_{\text{prior}}(|\mu_{\text{prior}} - \bar{\theta}_k^E| - \epsilon/2)}{\epsilon/2}$. We need $n_y^E \geq N_{\text{prior}}(\epsilon)$ (considering integer $n_y^E$).

**Bounding Term 2 (Data Estimation Error):** We require the contribution from this term to be less than $\epsilon/2$:

$$\frac{n_y^E}{S_{\text{prior}} + n_y^E}|\hat{\theta}_k^{(E)} - \bar{\theta}_k^E| < \frac{\epsilon}{2}$$

Since $\frac{n_y^E}{S_{\text{prior}} + n_y^E} < 1$ for $n_y^E > 0$, $S_{\text{prior}} \geq 2$, a sufficient condition is to ensure:

$$|\hat{\theta}_k^{(E)} - \bar{\theta}_k^E| < \frac{\epsilon}{2}$$

We want this inequality to hold with high probability. We use Hoeffding's inequality for the empirical mean of $n_y^E$ independent Bernoulli trials ($\hat{\theta}_k^{(E)}$) approximating the true mean ($\bar{\theta}_k^E$):

$$P(|\hat{\theta}_k^{(E)} - \bar{\theta}_k^E| \geq \eta) \leq 2e^{-2n_y^E \eta^2}$$

We want the probability of the undesired event ($|\hat{\theta}_k^{(E)} - \bar{\theta}_k^E| \geq \epsilon/2$) to be at most $\delta$. Setting $\eta = \epsilon/2$:

$$P(|\hat{\theta}_k^{(E)} - \bar{\theta}_k^E| \geq \epsilon/2) \leq 2e^{-2n_y^E(\epsilon/2)^2}$$

We require this probability to be less than or equal to $\delta$:

$$2e^{-2n_y^E(\epsilon/2)^2} \leq \delta$$

Solving for $n_y^E$:

$$e^{-2n_y^E(\epsilon/2)^2} \leq \frac{\delta}{2}$$

$$-2n_y^E(\epsilon/2)^2 \leq \ln\left(\frac{\delta}{2}\right)$$

$$2n_y^E(\epsilon/2)^2 \geq -\ln\left(\frac{\delta}{2}\right) = \ln\left(\frac{2}{\delta}\right)$$

$$n_y^E \geq \frac{\ln(2/\delta)}{2(\epsilon/2)^2} = \frac{2\ln(2/\delta)}{\epsilon^2}$$

Let $N_{\text{data}}(\epsilon, \delta) = \frac{2 \ln(2/\delta)}{\epsilon^2}$. We need $n_y^E \geq N_{\text{data}}(\epsilon, \delta)$.

**Combining Conditions:** To ensure that $|\mu_y^E - \bar{\theta}_k^E| < \epsilon$ with probability at least $1 - \delta$, we need to satisfy both the condition derived from bounding Term 1 (if applicable) and the condition derived from bounding Term 2 (probabilistically). Therefore, we require $n_y^E$ to be large enough for both:

$$n_y^E \geq \max \{N_{\text{prior}}(\epsilon), N_{\text{data}}(\epsilon, \delta)\}$$

Substituting the expressions for $N_{\text{prior}}(\epsilon)$ and $N_{\text{data}}(\epsilon, \delta)$ gives the result stated in the proposition. The first term is considered 0 if $|\mu_{\text{prior}} - \bar{\theta}_k^E| \leq \epsilon/2$. $\qquad\square$

### D.3    THEORETICAL CHARACTERISATIONS OF THE IFD DEFERRAL RULE

We analyse the asymptotic behaviour of the IFD loss function (Eq. (10)). We adopt the notation from §4.1, where $\bar{\theta}_y^E$ denotes the true underlying per-class accuracy of expert $E$, and $k_{\text{best}}^E = \arg\max_y \theta_y^E$ is the true best class.

**Proposition 4.3** (LCB Convergence). *Under the Beta–Binomial model of §4.1 with prior* $\text{Beta}(\alpha_y, \beta_y)$, *for each class* $y \in \mathcal{Y}$, *as the context count* $n_y^E \to \infty$, *the posterior mean* $\mu_y^E$ *converges almost surely to the true expert accuracy* $\bar{\theta}_y^E$, *and its variance* $(\sigma_y^E)^2 \to 0$. *Consequently, for any fixed* $\alpha \geq 0$, *the LCB weight* $L_y^E$ *converges almost surely to the true accuracy:*

$$L_y^E = [\, \mu_y^E - \alpha\, \sigma_y^E \,]_+ \xrightarrow{\text{a.s.}} \theta_y^E.$$

*Proof.* From §4.1, the posterior distribution for the accuracy parameter given context $D_C^E$ is $\text{Beta}(\alpha_y + t_y^E, \beta_y + n_y^E - t_y^E)$. The posterior mean is:

$$\mu_y^E = \frac{\alpha_y + t_y^E}{\alpha_y + \beta_y + n_y^E}.$$

By the Strong Law of Large Numbers (SLLN), the empirical accuracy $t_y^E/n_y^E \xrightarrow{\text{a.s.}} \theta_y^E$ as $n_y^E \to \infty$. We rewrite the posterior mean by dividing the numerator and denominator by $n_y^E$:

$$\mu_y^E = \frac{(\alpha_y/n_y^E) + (t_y^E/n_y^E)}{(\alpha_y + \beta_y)/n_y^E + 1}.$$

As $n_y^E \to \infty$, the terms involving the fixed prior parameters vanish (assuming finite $\alpha_y, \beta_y$). Thus, the posterior mean converges almost surely: $\mu_y^E \xrightarrow{\text{a.s.}} \frac{0 + \theta_y^E}{0 + 1} = \theta_y^E$.

The posterior variance is:

$$(\sigma_y^E)^2 = \frac{(\alpha_y + t_y^E)(\beta_y + n_y^E - t_y^E)}{(\alpha_y + \beta_y + n_y^E)^2(\alpha_y + \beta_y + n_y^E + 1)}.$$

As $n_y^E \to \infty$, the numerator is $O((n_y^E)^2)$ (since $t_y^E/n_y^E$ converges), while the denominator is $O((n_y^E)^3)$. Thus, the variance is $O(1/n_y^E)$ and converges to 0. Consequently, $\sigma_y^E \xrightarrow{\text{a.s.}} 0$.

Therefore, $L_y^E = [\, \mu_y^E - \alpha\sigma_y^E \,]_+$. By the continuous mapping theorem, $L_y^E \xrightarrow{\text{a.s.}} [\, \theta_y^E - \alpha \cdot 0 \,]_+ = \theta_y^E$ (since $\theta_y^E \in [0, 1]$). $\qquad\square$

**Theorem D.1** (Asymptotic IFD Rule Characterisation). *Let* $R_{IFD}(g)$ *be the population risk associated with the IFD loss* $\mathcal{L}_{CE}^{IFD}$ *(Eq. (10)). As the context counts* $n_y^E \to \infty$ *for all classes* $y$, *any minimiser* $g^*$ *of* $R_{IFD}$ *induces the following decision rule* $f^*(x, E)$:

$$f^*(x, E) = \begin{cases} \bot, & \text{if } \max_{y \in \mathcal{Y}} P(Y{=}y \mid X{=}x) \leq P(Y{=}k_{\text{best}}^E \mid X{=}x)\, \bar{\theta}_{k_{\text{best}}^E}^E \\ \arg\max_{y \in \mathcal{Y}} g_y^*(x), & \text{otherwise.} \end{cases}$$

*where* $k_{\text{best}}^E = \arg\max_y \theta_y^E$ *is the expert's true best class (assumed unique). Furthermore, the induced classifier* $h^*(x) = \arg\max_{y \in \mathcal{Y}} g_y^*(x)$ *is the Bayes-optimal multiclass predictor,* $h^*(x) = \arg\max_y P(Y{=}y \mid X{=}x)$.

*Proof.* We aim to minimise the population risk $R_{\text{IFD}}(g) = \mathbb{E}_{(X,Y,E)}[\mathcal{L}_{\text{CE}}^{\text{IFD}}(X,Y,E)]$. We proceed by minimising the conditional risk $R_{\text{IFD}}(g|x,E) = \mathbb{E}_{Y|X=x}[\mathcal{L}_{\text{CE}}^{\text{IFD}}(x,Y,E)]$ pointwise for each $(x,E)$. We assume $P(Y|X=x,E) = P(Y|X=x)$.

**1. Deriving the Limiting Conditional Risk.** The IFD loss (Eq. (10)) is:

$$\mathcal{L}_{\text{CE}}^{\text{IFD}}(x,y,E) \;=\; -\log p_y(x,E) \;-\; L_y^E \, \mathbb{I}\{\widehat{k}_{\text{best}}^E = y\} \, \log p_\perp(x,E).$$

Let $\eta_y(x) = P(Y=y|X=x)$. The conditional risk is:

$$R_{\text{IFD}}(g|x,E) = \sum_{y'\in\mathcal{Y}} \eta_{y'}(x) \left( -\log P_{y'}(x,E) - L_{y'}^E \, \mathbb{I}\{\widehat{k}_{\text{best}}^E = y'\} \log P_\perp(x,E) \right).$$

We consider the asymptotic limit where $n_y^E \to \infty$ for all $y$. In this limit:

- From Proposition 4.3, the LCB weight converges a.s. to the true accuracy, $L_{y'}^E \to \theta_{y'}^E$.
- From Proposition 4.1 (Peak-class identification), since $\mu_y^E \to \theta_y^E$ (by Prop. 4.3), the estimated best class $\widehat{k}_{\text{best}}^E = \arg\max_y \mu_y^E$ converges a.s. to the true best class $k_{\text{best}}^E = \arg\max_y \theta_y^E$.

Substituting these limits, the term $L_{y'}^E \mathbb{I}\{\widehat{k}_{\text{best}}^E = y'\}$ converges to $\theta_{y'}^E \mathbb{I}\{k_{\text{best}}^E = y'\}$. The limiting conditional risk $L(g|x,E)$ is:

$$L(g|x,E) = \sum_{y'\in\mathcal{Y}} \eta_{y'}(x) \left( -\log P_{y'}(x,E) - \theta_{y'}^E \mathbb{I}\{k_{\text{best}}^E = y'\} \log P_\perp(x,E) \right)$$

$$= \sum_{y'\in\mathcal{Y}} \eta_{y'}(x)[-\log P_{y'}(x,E)] - \log P_\perp(x,E) \sum_{y'\in\mathcal{Y}} \eta_{y'}(x)\theta_{y'}^E \mathbb{I}\{k_{\text{best}}^E = y'\}$$

The second sum simplifies because the indicator is non-zero only when $y' = k_{\text{best}}^E$. Thus, the sum becomes $\eta_{k_{\text{best}}^E}(x) \bar{\theta}_{k_{\text{best}}^E}^E$. The limiting conditional risk is:

$$L(g|x,E) = - \sum_{y'\in\mathcal{Y}} \eta_{y'}(x) \log P_{y'}(x,E) - \left( \eta_{k_{\text{best}}^E}(x) \bar{\theta}_{k_{\text{best}}^E}^E \right) \log P_\perp(x,E).$$

**2. Deriving Optimal Model Probabilities.** The risk $L(g|x,E)$ is a weighted cross-entropy loss for the $(K+1)$ categories $\mathcal{Y} \cup \{\perp\}$. The weights are $w_y = \eta_y(x)$ for $y \in \mathcal{Y}$, and $w_\perp = \eta_{k^\star}(x)\theta_{k^\star}^E$. The total weight is $W = \sum_y w_y + w_\perp = 1 + w_\perp$. The minimising probabilities $P^*(x,E)$ are proportional to these weights (see e.g., Mozannar & Sontag (2021)):

$$p_y^*(x,E) = \frac{w_y}{W} = \frac{\eta_y(x)}{1 + \eta_{k_{\text{best}}^E}(x) \bar{\theta}_{k_{\text{best}}^E}^E} \quad \text{for } y \in \mathcal{Y},$$

$$p_\perp^*(x,E) = \frac{w_\perp}{W} = \frac{\eta_{k_{\text{best}}^E}(x) \bar{\theta}_{k_{\text{best}}^E}^E}{1 + \eta_{k_{\text{best}}^E}(x) \bar{\theta}_{k_{\text{best}}^E}^E}.$$

**3. Induced Classifier is Bayes-Optimal.** The classifier $h^*(x)$ maximises $g_y^*(x)$, which (due to softmax monotonicity) is equivalent to maximizing $p_y^*(x,E)$. Since the denominator $W$ is constant for a fixed $(x,E)$, maximizing $p_y^*(x,E)$ is equivalent to maximizing $\eta_y(x)$. Therefore, $h^*(x) = \arg\max_{y\in\mathcal{Y}} \eta_y(x) = \arg\max_{y\in\mathcal{Y}} P(Y=y|X=x)$, the Bayes-optimal classifier.

**4. Deferral Rule.** The system defers if $g_\perp^*(x,E) \geq \max_{y\in\mathcal{Y}} g_y^*(x)$, which is equivalent to $p_\perp^*(x,E) \geq \max_{y\in\mathcal{Y}} p_y^*(x,E)$. Substituting the optimal probabilities and canceling the positive denominator $W$:

$$\eta_{k_{\text{best}}^E}(x) \bar{\theta}_{k_{\text{best}}^E}^E \geq \max_{y\in\mathcal{Y}} \eta_y(x).$$

Substituting back $\eta_y(x) = P(Y=y|X=x)$ yields the deferral condition stated in the theorem. $\qquad\square$

**Remark D.1** (Interpretation of the Asymptotic IFD Rule). *Theorem D.1 shows that IFD asymptotically implements the Bayes classifier $h^*(x)$. However, its deferral rule differs from the standard L2D Bayes-optimal rule derived from the 0—1 loss (Eq. (6)). The standard rule defers if $\max_y P(Y=y|x) \leq P(Y=M|x,E)$.*

*Assuming the class-conditional expert correctness model (i.e., $P(M{=}y \mid Y{=}y, E) \approx \theta_y^E$) and relevant conditional independencies, the standard rule compares model confidence to the expert's expected accuracy:*

$$\max_y P(Y{=}y \mid X{=}x) \le \sum_y P(Y{=}y \mid X{=}x)\, \theta_y^E = \mathbb{E}_{Y|x}[\theta_Y^E].$$

*The IFD rule, in contrast, defers if:*

$$\max_y P(Y{=}y \mid X{=}x) \le P(Y{=}k_{\text{best}}^E \mid X{=}x)\, \bar{\theta}_{k_{\text{best}}^E}^E.$$

*This distinct rule arises from the structure of the $\mathcal{L}_{CE}^{IFD}$ surrogate (Eq. (10)). The deferral term rewards deferral only when the true class $y$ aligns with the expert's peak class $\widehat{k}_{\text{best}}^E$ (asymptotically $k_{\text{best}}^E$). This encourages deferral based on the expert's* peak competence, *rather than their expected correctness across all likely classes.*

# E PERMUTATION SYMMETRY, INVARIANCE, AND GENERALISATION OF IDENTITY-FREE DEFERRAL

## E.1 SETUP, NOTATION, AND PROBLEM FORMULATION

**Labels, inputs, experts.** We consider a $K$-class classification problem with label set $\mathcal{Y} = \{1, \dots, K\}$, inputs $X \in \mathcal{X}$, and target $Y \in \mathcal{Y}$. Let $(X, Y) \sim \mathcal{P}$. For any $x$, denote the posterior

$$\eta_y(x) \doteq P(Y = y \mid X = x), \qquad \eta(x) = (\eta_1(x), \dots, \eta_K(x)).$$

An expert $E$ provides a discrete prediction $M^E \in \mathcal{Y}$.

Each expert $E$ has a context set

$$D_C^E = \{(x_i^C, y_i^C, m_i^{E,C})\}_{i=1}^{N_C^E}, \qquad m_i^{E,C} \in \mathcal{Y},$$

summarising past behaviour. An encoder $\psi$ may map this context to a representation; we will write the deferral logit as $g_\perp(x, E)$.

**Classifier and logits.** Let $g_y(x)$, $y \in \mathcal{Y}$, be the class logits used by the classifier $h$,

$$h(x) = \arg\max_{y \in \mathcal{Y}} g_y(x) \quad \text{(with any fixed measurable tie-breaker if needed)}.$$

**Rejector (deferral rejector) and $(K{+}1)$ softmax.** The rejector produces a single deferral logit $g_\perp(x, E) \in \mathbb{R}$. Define the joint logits and normaliser

$$\tilde{g}(x, E) = \big(g_1(x), \dots, g_K(x),\, g_\perp(x, E)\big), \qquad Z(x, E) = \exp\{g_\perp(x, E)\} + \sum_{y \in \mathcal{Y}} \exp\{g_y(x)\}.$$

The $(K{+}1)$-way probabilities are

$$p(Y = y \mid x, E) = \frac{\exp\{g_y(x)\}}{Z(x, E)} \quad (y \in \mathcal{Y}), \qquad p(\perp \mid x, E) = \frac{\exp\{g_\perp(x, E)\}}{Z(x, E)}.$$

At inference, the system predicts $h(x)$ and defers according to the rejector

$$r(x, E) = \mathbb{I}\left[ \max_{y \in \mathcal{Y}} g_y(x) \le g_\perp(x, E) \right],$$

so the overall predictor is $\hat{Y}(x, E) = h(x)$ if $r(x, E) = 0$ and $\hat{Y}(x, E) = \perp$ if $r(x, E) = 1$.

**Query-conditioned expert correctness and $q(x, E)$.** Define the expert's query-conditioned correctness

$$q(x, E) \doteq P(M^E = Y \mid X = x, E).$$

Define the class-conditional correctness profile at query $x$,

$$\mu_y^E(x) \doteq P(M^E = y \mid X = x, Y = y, E) \in [0, 1],$$

so that by the law of total probability

$$q(x, E) \ = \ \sum_{y=1}^{K} P(Y = y \mid X = x, \ E) \, \mu_y^E(x).$$

When additionally $Y \perp E \mid X$, we have $q(x, E) = \sum_{y=1}^{K} \eta_y(x) \, \mu_y^E(x)$.

**0—1 system loss and Bayes choice.** Under the 0—1 system loss, the pointwise risks at $(x, E)$ are

$$R_{\text{predict}}(x) \ = \ 1 - \max_{y \in \mathcal{Y}} P(Y = y \mid X = x) \qquad \text{and} \qquad R_{\text{defer}}(x, E) \ = \ 1 - q(x, E).$$

Equivalently, using $h(x) = \arg\max_y g_y(x)$,

$$R_{\text{predict}}(x) \ = \ 1 - \eta_{h(x)}(x).$$

Define the (system) deferral margin

$$\Delta_{\text{sys}}(x, E) \ \doteq \ q(x, E) \ - \ \max_{y \in \mathcal{Y}} P(Y = y \mid X = x) \ = \ q(x, E) \ - \ \eta_{h(x)}(x).$$

The Bayes action is to defer iff $\Delta_{\text{sys}}(x, E) > 0$:

$$r^\star(x, E) \ = \ \mathbb{I}\left[ P(Y = M \mid X = x, E) \ \geq \ \max_{y \in \mathcal{Y}} P(Y = y \mid X = x) \right].$$

**A regret proxy for a given rejector.** Given logits $\tilde{g} = (g_1, \ldots, g_K, g_\perp)$, define the induced action

$$a_{\tilde{g}}(x, E) = \begin{cases} \text{defer}, & g_\perp(x, E) \ \geq \ \max_y g_y(x), \\ \text{predict}, & \text{otherwise}, \end{cases}$$

and the pointwise regret

$$\text{reg}(\tilde{g}; x, E) \ = \ \max(0, \Delta_{\text{sys}}(x, E)) \, \mathbf{1}\{a_{\tilde{g}} = \text{predict}\} \ + \ \max(0, -\Delta_{\text{sys}}(x, E)) \, \mathbf{1}\{a_{\tilde{g}} = \text{defer}\}.$$

This is zero iff the rejector matches the Bayes decision at $(x, E)$.

**Coherent relabelling (permutation symmetry).** Let $\mathfrak{S}_K$ be the permutation group on $\{1, \ldots, K\}$ and define its action on any class-indexed quantity $v = \{v_y\}_{y=1}^{K}$ by

$$(v^\pi)_y \ \coloneqq \ v_{\pi^{-1}(y)}.$$

A *coherent relabelling* applies $\pi$ to all class-indexed objects:

$$(M, Y) \mapsto \big(\pi(M), \pi(Y)\big), \qquad P^\pi(Y{=}y \mid X{=}x) \ \coloneqq \ P(Y{=}\pi^{-1}(y) \mid X{=}x),$$

and to the context

$$D_C^{E, \pi} \ = \ \{(x_i^C, \ \pi(y_i^C), \ \pi(m_i^{E, C}))\}_{i=1}^{N_C^E}.$$

Because the event $\{M{=}Y\}$ is invariant under $(M, Y) \mapsto (\pi(M), \pi(Y))$, we have $P(Y{=}M \mid X{=}x, E)$ unchanged, and $\max_y P(Y{=}y \mid X{=}x)$ unchanged (the $\arg\max$ simply relabels).

**OOD experts via identity shifts.** We model an "expert with the same competence structure under a new label identity" by permuting class-indexed expert profiles while leaving the task fixed. For class-conditional correctness, this amounts to

$$\mu_y^{E^\pi}(x) \ \coloneqq \ \mu_{\pi^{-1}(y)}^E(x) \quad \implies \quad q(x, E^\pi) \ = \ \sum_{y=1}^{K} P(Y{=}y \mid X{=}x, E^\pi) \, \mu_{\pi^{-1}(y)}^E(x).$$

Under (A1) $Y \perp E \mid X$, this reduces to $q(x, E) \ = \ \sum_y \eta_y(x) \mu_y^E(x)$ and $q(x, E^\pi) \ = \ \sum_y \eta_y(x) \mu_{\pi^{-1}(y)}^E(x)$. Expert-only permutations can therefore *change* the Bayes action when they alter $q(x, E)$ relative to $\max_y P(Y = y \mid X = x)$. This motivates permutation-based OOD stress tests used later.

## E.2 WHY L2D-POP FAILS ON OOD EXPERTS: QUERY-INDEPENDENT VARIANT

**Architecture.** The query-independent (QI) variant encodes the expert's context via a Deep Sets Zaheer et al. (2018) mean:

$$\psi_{\text{QI}}^E(D_C^E;\theta) \;=\; \frac{1}{N_C^E}\sum_{i=1}^{N_C^E}\gamma\big(\,[\varphi(x_i^C);\;e(y_i^C);\;e(m_i^{E,C})]\,;\theta\big),$$

where $\varphi(\cdot)$ is a fixed feature map and $e(\cdot)$ is either a one-hot or a learned class embedding.[2] The deferral logit is

$$g_\perp^{\text{Pop-ind}}(x,E) = r_\phi\Big(\,[\psi_{\text{QI}}^E(D_C^E;\theta);\;\varphi(x)]\,\Big).$$

**Set-permutation vs. coherent relabelling invariance.** Deep Sets confers invariance to *set permutations*: reordering the elements of $D_C^E$ leaves $\psi_{\text{QI}}^E$ unchanged. The symmetry relevant to label identity is *coherent relabelling* (§3.3): applying the same permutation $\pi \in \mathfrak{S}_K$ to all class-indexed objects in each token, $(y,m) \mapsto (\pi(y),\pi(m))$. Deep Sets does *not* guarantee invariance to such label relabellings.

**Lemma E.1** (Generic non-invariance of a token MLP with class-specific columns). *Let the token be $t(y,m) = [\varphi(x^C);\;e(y);\;e(m)] \in \mathbb{R}^{d+2K}$ (with one-hot $e(\cdot)$; the learned-embedding case is covered in footnote 2). Suppose the first affine layer of $\gamma$ is*

$$z \;=\; W_x\,\varphi(x^C) \;+\; W_y\,e(y) \;+\; W_m\,e(m) \;+\; b, \qquad W_y,W_m \in \mathbb{R}^{H\times K}.$$

*If not all columns of $W_y$ are identical and not all columns of $W_m$ are identical, then* generically *(i.e., for an open set of downstream parameters) there exist $(x^C,y,m,\pi)$ such that*

$$\gamma\big([\varphi(x^C);P_\pi e(y);P_\pi e(m)]\big) \;\neq\; \gamma\big([\varphi(x^C);e(y);e(m)]\big).$$

**Proposition 4.3** (QI encoder is generically non-invariant under coherent relabelling). *Let*

$$\psi_{\text{QI}}^E(D_C^E;\theta) \;=\; \frac{1}{N_C^E}\sum_{(x^C,y,m)\in D_C^E}\gamma\big([\varphi(x^C);\;e(y);\;e(m)];\theta\big).$$

*If there exist $\pi \in \mathfrak{S}_K$ and a token $(x^C,y,m)$ such that $\gamma([\varphi(x^C);P_\pi e(y);P_\pi e(m)];\theta) \neq \gamma([\varphi(x^C);e(y);e(m)];\theta)$, then there exist a one-element context $D_C^E = \{(x^C,y,m)\}$ for which*

$$\psi_{\text{QI}}^E(D_C^{E,\pi};\theta) \;\neq\; \psi_{\text{QI}}^E(D_C^E;\theta).$$

*In particular, for standard L2D-Pop with untied class embeddings and an unconstrained MLP $\gamma$, this non-invariance holds* generically.

**Corollary E.1** (QI Rejector inherits non-invariance generically). *If $r_\phi$ is non-constant, then* generically *there exist $(x,E)$ and a permutation $\pi$ such that*

$$g_\perp^{\text{Pop-ind}}(x,E^\pi) \;\neq\; g_\perp^{\text{Pop-ind}}(x,E).$$

**Concrete example** ($K{=}2$). Let $e(1) = (1,0)^\top$, $e(2) = (0,1)^\top$ and suppose $\gamma$ is linear in the class channels:

$$\gamma\big([\varphi(x^C);\;e(y);\;e(m)]\big) \;=\; A\,e(y) \;+\; B\,e(m) \;+\; u^\top\varphi(x^C),$$

with $A,B \in \mathbb{R}^{d\times 2}$ and $u \in \mathbb{R}^{\dim\varphi}$. Under the swap $\pi = (1\ 2)$, the token output shifts by $(A{+}B)\,(e(2)-e(1))$, which is nonzero except on a measure-zero parameter set. The claims follow by Prop. 4.3 and Cor. E.1.

**Consequences for OOD risk.** Let $\Pi$ be any distribution over permutations of labels and let $\ell$ be 1-Lipschitz in $g_\perp$ (classifier logits fixed). Then

$$\mathcal{L}_{\text{OOD}}(g_\perp) - \mathcal{L}_{\text{ID}}(g_\perp) \;\leq\; \mathbb{E}_{(X,E,D_C^E)}\big[\,\mathbb{E}_{\pi\sim\Pi}\big|\,g_\perp(X,E^\pi) - g_\perp(X,E)\,\big|\,\big],$$

and the inner expectation is *strictly positive generically*.

**generalisation viewpoint.** Let $\mathcal{H}_{\text{QI}}$ be the induced hypothesis class. Because $\mathcal{H}_{\text{QI}}$ contains identity-conditioned functions, it effectively contains $K!$ distinct "cases" for a single structural pattern (one per permutation). With finite ID coverage of permutations, ERM can overfit to the seen identities. A permutation-invariant rejector collapses these $K!$ cases to one.

---

[2] For learned embeddings, write $e(y) = E^\top e_y$ with $E \in \mathbb{R}^{K\times d_e}$ and $e_y$ the $y$th canonical basis vector. All arguments below apply verbatim to the effective linear map multiplying $e(y)$.

### E.3 WHY L2D-POP FAILS ON OOD EXPERTS: QUERY-CONDITIONED VARIANT

**Architecture.** The query-conditioned (QC) variant builds a query from the current input and cross-attends to the expert's context:

$$q(x) = W_Q \, \varphi(x), \quad h_i = \gamma\big([\varphi(x_i^C); \, e(y_i^C); \, e(m_i^{E,C})]; \theta\big), \quad k_i = W_K \, \varphi(x_i^C), \quad v_i = W_V \, h_i,$$

$$\alpha_i(x) = \text{softmax}_i\big(q(x)^\top k_i / \sqrt{d}\big), \qquad s^E(x) = \text{FFN}\Big(\text{LN}\big(q(x) + \sum_i \alpha_i(x) \, v_i\big)\Big)$$

with $g_\perp^{\text{Pop-cond}}(x, E) = r_\phi([\varphi(x); \, s^E(x)])$. As above, $\varphi$ is a fixed feature map and $e(\cdot)$ denotes a class-ID embedding (one-hot or learned).

**Set-permutation vs. coherent label relabelling.** Without positional encodings, the cross-attention is invariant to *context index* permutations (reordering the items of $D_C^E$). The OOD symmetry is *coherent relabelling* (§3.3): $(y_i^C, m_i^{E,C}) \mapsto (\pi(y_i^C), \pi(m_i^{E,C}))$ for a single $\pi \in \mathfrak{S}_K$ applied to all tokens. Attention alone does not enforce invariance to such relabellings.

**Proposition 4.3** (QC cross-attention is generically non-invariant under coherent relabelling). *Let $D_C^{E,\pi}$ be obtained by applying $\pi \in \mathfrak{S}_K$ to all class-indexed fields of $D_C^E$. Assume the token map $\gamma$ is not $S_K$-equivariant (Lemma E.1). Then, for an open set of parameters $(W_V, \text{FFN}, \text{LN}, r_\phi)$ and for some $(x, E, \pi)$,*

$$s^{E,\pi}(x) \neq s^E(x) \quad \text{and} \quad g_\perp^{\text{Pop-cond}}(x, E^\pi) \neq g_\perp^{\text{Pop-cond}}(x, E).$$

**Amplified shortcut via query/identity interaction.** Compared to QI, QC can learn *bilinear, index-aligned* features that explicitly tie the current *query class* to the same *absolute* index in the context. A single head can implement

$$g_\perp^{\text{Pop-cond}}(x, E) \approx \sum_{j=1}^K a_j \, \rho_j(x) \underbrace{\phi_j(D_C^E)}_{\text{``evidence for class } j\text{''}},$$

with untied coefficients $a_j$ and class-indexed context statistics $\phi_j$, strengthening identity conditioning.

**Concrete counterexample (index alignment, $K{=}3$).** Suppose $\gamma$ produces a vector whose $j$th coordinate encodes "specialisation evidence for class $j$. Choose $W_Q, W_K$ so that $q(x)^\top k_i$ is large when $x_i^C$ resembles $x$ for the model's top class $k_{\text{top}}(x)$, and choose $W_V, \text{FFN}, r_\phi$ so that $g_\perp(x, E) \approx \sum_{j=1}^3 a_j \, \rho_j(x)$ [evidence for class $j$]. If an OOD expert $E'$ presents the same specialisation pattern on index $\pi(j)$, the values do not transform by $P_\pi$, the weights $\alpha_i$ remain fixed, and $g_\perp$ fails to implement the coherent relabelling.

**Consequences for OOD risk (comparison with QI).** With the same Lipschitz and permutation averaging setup as in §E.2,

$$\mathcal{L}_{\text{OOD}}(g_\perp) - \mathcal{L}_{\text{ID}}(g_\perp) \leq \mathbb{E}_{(X,E,D_C^E)}\big[\mathbb{E}_{\pi \sim \Pi}\big|g_\perp(X, E^\pi) - g_\perp(X, E)\big|\big],$$

and the inner expectation is *strictly positive generically* (Prop. 4.3). The bilinear index alignment expands $\mathcal{H}_{\text{QC}}$, typically increasing the OOD penalty relative to QI.

**Remark (positional encodings).** If positional encodings over the context sequence are used, even set-index permutation invariance is lost; this does not mitigate—and can exacerbate—the label-relabelling non-invariance above.

### E.4 IFD: ARCHITECTURE AND INVARIANCE PROOF

**Setup and notation.** Let $g(x) = (g_1(x), \ldots, g_K(x))$ be the classifier logits and $\rho(x) = \text{softmax}(g(x))$ the class–probability vector with coordinates $\rho_k(x)$. From the expert's context $D_C^E$, fix $J \geq 1$ per-class posterior functionals $\mathcal{F} = \{f_j\}_{j=1}^J$ and define

$$f_j^E(y) := f_j(\theta_y^E \mid D_C^E), \qquad \Phi^E := [f_j^E(y)]_{y=1..K, \, j=1..J} \in \mathbb{R}^{K \times J}.$$

Choose a ranking functional $u \in \mathcal{F}$ and let $u^E \in \mathbb{R}^K$ denote its per-class values, $u_y^E := u(\theta_y^E \mid D_C^E)$. We use a fixed, permutation–equivariant tie–breaker shared across the paper. Write $k_{\text{top}}(x) \in \arg\max_k \rho_k(x)$ and $\widehat{k}_{\text{best}}^E \in \arg\max_y u_y^E$. For $\pi \in \mathfrak{S}_K$, the induced action on any class–indexed vector $v$ is $(v^\pi)_y := v_{\pi^{-1}(y)}$, and on the functional matrix is

$$(\Phi^{E,\pi})_{y,j} = (\Phi^E)_{\pi^{-1}(y),j} \iff \Phi^{E,\pi} = P_\pi \Phi^E,$$

which also yields $(u^{E,\pi})_y = u_{\pi^{-1}(y)}^E$.

**Role indexing and identity–free state.** The identity–free state $z(x,E) \in \mathbb{R}^{d_z}$ is any fixed–dimension vector built by *reading values at roles* and *aggregating symmetric statistics*, never exposing absolute class IDs:

$$z(x,E) = \text{RI}(\rho(x), \Phi^E). \tag{20}$$

Concretely, $\text{RI}(\cdot)$ may include: (i) *role values* such as $\rho_{k_{\text{top}}(x)}(x)$, $u_{k_{\text{top}}(x)}^E$, $\rho_{\widehat{k}_{\text{best}}^E}(x)$, $u_{\widehat{k}_{\text{best}}^E}^E$, or any other $f_j^E$ evaluated at these roles; (ii) *role comparisons* such as $\mathbb{I}[\, k_{\text{top}}(x){=}\widehat{k}_{\text{best}}^E\,]$, gaps $\rho_{k_{\text{top}}(x)}(x){-}\rho_{(2)}(x)$ and $u_{(1)}^E - u_{(2)}^E$; and (iii) *symmetric aggregates* of $\rho$ (e.g., entropy $H(\rho)$) or of rows of $\Phi^E$ (e.g., norms), as long as they are permutation-invariant. A concrete 6–dimensional choice used in our experiments (with $J{=}2$, $f_1 = $ mean and $f_2 = $ variance, $u \equiv$ mean) is

$$z(x,E) = \big(\rho_{k_{\text{top}}(x)}(x), \rho_{\widehat{k}_{\text{best}}^E}(x), \mu_{k_{\text{top}}(x)}^E, (\sigma_{k_{\text{top}}(x)}^E)^2, \mu_{\widehat{k}_{\text{best}}^E}^E, (\sigma_{\widehat{k}_{\text{best}}^E}^E)^2\big). \tag{21}$$

The IFD rejector applies a small MLP to $z$:

$$g_\perp^{\text{IFD}}(x,E) = r_\phi\big(z(x,E)\big), \tag{22}$$

and the $(K{+}1)$–way softmax over $\mathcal{Y} \cup \{\perp\}$ defines the deferral probability

$$\rho_\perp(x,E) := \text{softmax}\big(g_1(x), \ldots, g_K(x), g_\perp^{\text{IFD}}(x,E)\big)_\perp.$$

**Permutation action and role equivariance.** Under a coherent relabelling $\pi \in \mathfrak{S}_K$ acting on all class–indexed quantities, $(\rho^\pi)_y = \rho_{\pi^{-1}(y)}$ and $(\Phi^{E,\pi})_{y,j} = (\Phi^E)_{\pi^{-1}(y),j}$. By definition of $\arg\max$ and the fixed tie–breaker, the role indices transform *equivariantly*:

$$k_{\text{top}}(\rho^\pi) = \pi\big(k_{\text{top}}(\rho)\big), \qquad \widehat{k}_{\text{best}}^E(\Phi^{E,\pi}) = \pi\big(\widehat{k}_{\text{best}}^E(\Phi^E)\big). \tag{23}$$

Consequently, values read *at roles* are preserved; e.g., $(\rho^\pi)_{\pi(k_{\text{top}}(\rho))} = \rho_{k_{\text{top}}(\rho)}$ and $u_{\pi(k_{\text{top}}(\rho))}^{E,\pi} = u_{k_{\text{top}}(\rho)}^E$.

**Proof of Proposition 4.2 (permutation invariance of the role-indexed state).** We must show that $z$ is invariant under coherent relabelling and, hence, so is $g_\perp^{\text{IFD}}$ by composition with $r_\phi$. Each coordinate of $z$ belongs to one of three classes:

- *Role values* (e.g., $\rho_{k_{\text{top}}(\rho)}$, $u_{k_{\text{top}}(\rho)}^E$, $\rho_{\widehat{k}_{\text{best}}^E}$, $u_{\widehat{k}_{\text{best}}^E}^E$, or any $f_j^E$ at these roles): by (23), the index moves to $\pi(\cdot)$ while the value at that index is unchanged when the underlying vector/matrix is permuted coherently. Hence every role value is preserved.

- *Symmetric aggregates* (e.g., $H(\rho)$, norms or fixed-order statistics with a label-invariant tie-breaker): these are permutation–invariant by construction.

- *Role comparisons* (e.g., $\mathbb{I}[\, k_{\text{top}}(\rho){=}\widehat{k}_{\text{best}}^E\,]$, gaps between role values): since both roles transform by the same $\pi$, equalities and differences are preserved.

Therefore,

$$\text{RI}\big(\rho^\pi(x), \Phi^{E,\pi}\big) = \text{RI}\big(\rho(x), \Phi^E\big) \quad \text{for all } \pi \in \mathfrak{S}_K,$$

and invariance of the rejector follows:

$$g_\perp^{\text{IFD}}(x, E^\pi) = r_\phi\big(\text{RI}(\rho^\pi, \Phi^{E,\pi})\big) = r_\phi\big(\text{RI}(\rho, \Phi^E)\big) = g_\perp^{\text{IFD}}(x, E).$$

$\square$

**Corollary (rejector invariance).** For any parameters $\phi$ and any $\pi \in \mathfrak{S}_K$, $g_\perp^{\text{IFD}}(x, E^\pi) = g_\perp^{\text{IFD}}(x, E)$. Hence the IFD deferral decision is invariant to coherent relabelling.

**Scope and caveats.** (1) The invariance proven here is with respect to *coherent* relabellings acting on all class–indexed quantities $(\rho, \Phi^E)$. It *does not* assert invariance under *expert–only* permutations with fixed classifier/posterior; such permutations can change the Bayes action (Prop. 3.2). (2) The role selection uses $\arg\max$, which is non–differentiable only on measure–zero tie sets; the fixed tie–breaker makes $z$ measurable and the rejector trainable almost everywhere. (3) Invariance holds irrespective of calibration quality; it is enforced *architecturally* by the construction of $z$.

**Why this proof matters for OOD experts.** OOD experts often instantiate the *same* competence structure under different absolute class names. Because IFD collapses all $K!$ relabellings of a structure to the *same* $z(x, E)$, a single learned rule covers all permutations. In contrast, population encoders with identity channels must effectively relearn across permutations unless explicitly symmetrized; see the non–invariance results for L2D-Pop in App. E.2–E.3.

### E.5 PLAIN-LANGUAGE TAKEAWAYS

- **The symmetry that matters** (§E.1) is *coherent relabelling*: if we consistently rename classes everywhere (model outputs, expert profile, labels), the underlying decision problem is the same. IFD respects this by construction; generic L2D-Pop encoders do not.

- **Why L2D-Pop fails OOD** (§E.2–§E.3): its encoders expose *label–identity channels*. That lets the rejector learn shortcuts like "index 3 is special," or more subtly "defer when the model currently favours index $k$ *and* the context also looks strong at that same absolute index." These identity-keyed rules break when the same competence structure shows up under different indices OOD. In other words, the hypothesis class is not closed under permutations, so the learned function can change when you just rename classes.

- **Why IFD succeeds OOD** (§E.4): it first turns per-class signals into a small set of *role-indexed values* $z$ (e.g., "model's top probability," "expert's competence at the model's top class," and symmetric aggregates like an inner product), then *drops the indices*. We showed $z$ (and hence the rejector) is invariant to coherent relabelling, so a single learned rule automatically covers all $K!$ ways the same structure can be labelled.

## F EXPERIMENTAL DETAILS

**Computational Resources** All experiments in this paper were run using an NVIDIA GeForce RTX 4090 GPU, AMD Ryzen 9 7950X 16-Core Processor and 64GB DDR5 RAM.

**Classifier Models** The classifier architecture varies by dataset: ResNet-34 (He et al., 2015) for **HAM10000**, and a pretrained EfficientNet B0 (Tan & Le, 2020) for `Liver tumours`, and MobileNetV2 (Sandler et al., 2019) for `Blood Cells`.

**Rejector Model** For all experimental models, we keep the architecture of the rejector constant. For this, we utilise a 6-layer MLP with hidden dimensionality of 256.

**Training Details** Details on number of ID/OOD experts, context set sizes $N_C^E$ and $n_C^E$ are provided in Table 1. We utilise training and validation batch sizes of 128 for all datasets. We utilise the Adam optimiser (**?**) for all experiments, with a learning rate of 0.001 for `Liver tumours` and **HAM10000**, and 0.0005 for `Blood Cells`. All models are trained until convergence, measured via early stopping on the validation set measuring AURSAC (patience 50). We utilise a standard $\alpha = 1$ hyperparameter for the LCB-weighted loss for all experiments.

## G ADDITIONAL EXPERIMENTS

### G.1 TOY EXAMPLE: IDENTITY LEAKAGE VS. INVARIANCE

To illustrate the identity-conditioned failure mode discussed in §3.3 and the robustness of IFD, we design a simple 2D, 3-class ($K=3$) toy experiment.

DATA AND EXPERT SETUP

**Data.** We generate a 2D, 3-class dataset ($N=1000$) composed of three concentric circles: Class 0 forms the inner circle, Class 1 the middle annulus, and Class 2 the outer annulus (Figure 1). Moderate base noise (0.7) and additive Gaussian noise ($\sigma=0.5$) create strong overlap between neighbouring classes. A small MLP classifier is trained on this data and used as a fixed feature extractor for all models.

**Experimental controls.** Both pipelines (L2D-Pop and IFD) use the **same pretrained MLP feature extractor**, identical training procedure, random seeds, and context-set sampling. The only difference lies in the rejector architecture and how expert context is summarised.

**Expert competence.** During training, both models observe context sets from a single **Class 0 specialist**: high accuracy (90%) on Class 0, and low accuracy (50%) on Classes 1 and 2. At test time, we evaluate under two experts:

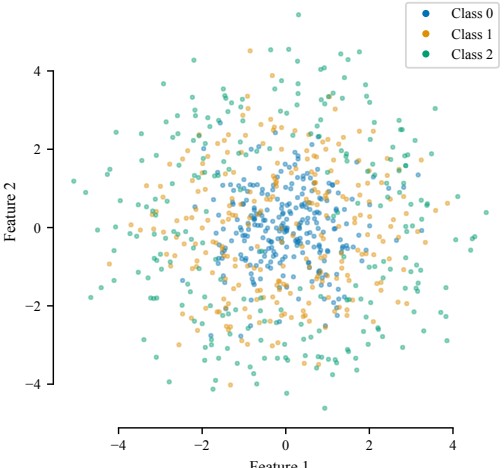

Figure 1: Toy dataset. Each colour corresponds to a class (three concentric rings).

- **ID Expert:** another Class 0 specialist (in-distribution).
- **OOD Expert:** a **Class 2 specialist** with the same competence profile permuted (high on Class 2, low on 0 and 1). This is an *expert-only permutation*—the task is fixed, but the expert's strengths are reindexed.

**Hypothesis.** Because L2D-Pop encodes absolute label identities, it is expected to learn an *identity-conditioned* rule (e.g., "defer on Class 0"), and therefore fail to adapt when the expert's specialty moves to Class 2. IFD, which removes label-identity channels and summarises per-class competence symmetrically, should instead learn a structural rule (e.g., "defer on whichever class the expert is best at") and transfer correctly under the expert-only permutation.

RESULTS AND ANALYSIS

Figure 2 visualises the deferral patterns determined by $g_\perp(x, \psi) \geq \max_y g_y(x)$. Points deferred to the expert are labelled by their true class; non-deferred points are shown as faint background dots.

**Interpretation.** L2D-Pop's deferral pattern is largely fixed: it continues to defer heavily on Classes 0–1 even after the expert's competence shifts to Class 2. This demonstrates an *identity-conditioned shortcut* consistent with Thm. 3.1. IFD, in contrast, changes its deferral behaviour automatically—defer decisions migrate from Class 0 (ID) to Class 2 (OOD)—showing the expected invariance predicted by Prop. 4.2.

G.2 ABLATION: UNCERTAINTY-AWARE LCB WEIGHTING

**Setup.** To isolate the effect of the uncertainty-aware LCB weighting in the IFD loss (Eq. (10)), we ablated the deferral weighting on three datasets: HAM10000, Blood Cells, and Liver tumours (as seen in §5). We compare four variants, averaged over five seeds, and *only* report the AURSAC and AURDAC across the full deferral budget:

1. **Full** (ours): $L_y^E = \left[ \mu_y^E - \alpha \, \sigma_y^E \right]_+$ with $\alpha = 1.0$.
2. **Mean**: $L_y^E = \mu_y^E$ (equivalent to $\alpha = 0$).
3. **Binary**: $L_y^E = \mathbb{1}\{y = \widehat{k}_{\text{best}}^E\}$.
4. **Classifier**: no deferral (for reference).

All other details (data preprocessing, model backbones, training schedules, and context construction) follow the main experiments. We aggregate system performance across experts (no per-expert reporting here).

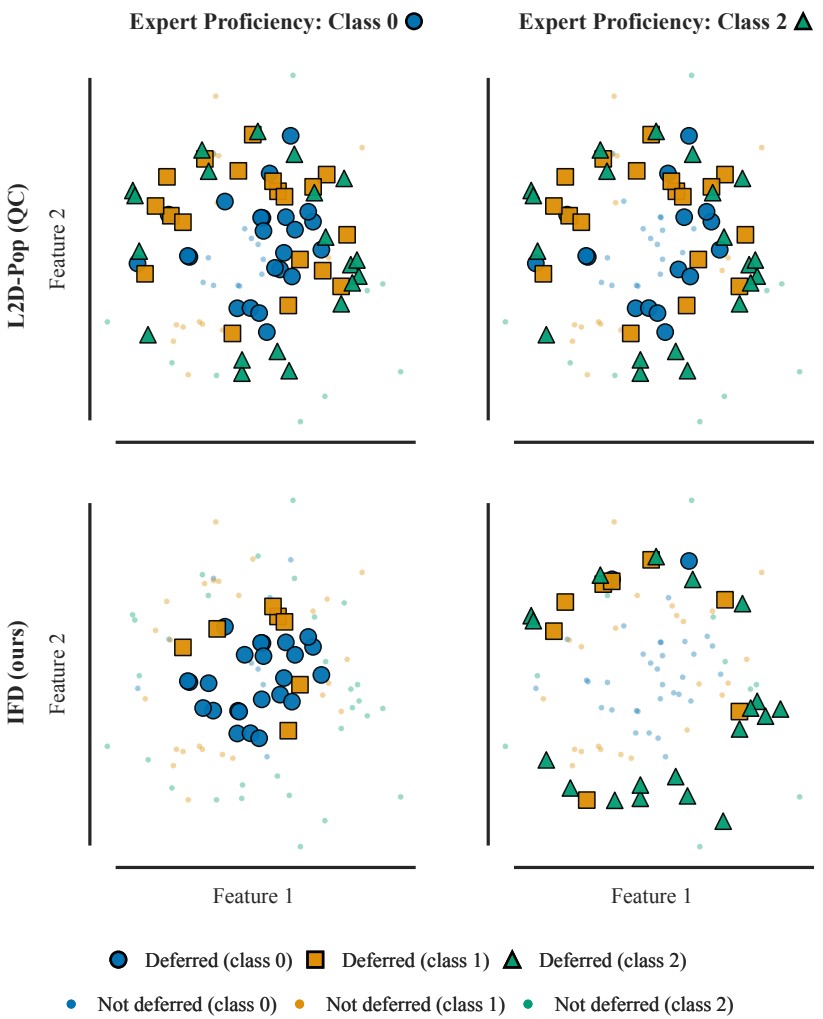

Figure 2: Deferral behaviour on the toy test set. **Top:** L2D-Pop; **Bottom:** IFD. **Left:** ID Expert (Class 0 specialist). **Right:** OOD Expert (Class 2 specialist, an expert-only permutation). L2D-Pop fails to adapt, whereas IFD correctly shifts deferrals to the new expert's specialty. Deferred points are determined by the default L2D deferral operation $g_\perp(x, \psi) \geq \max_y g_y(x)$.

**Findings.** Across datasets, **Full** is best or statistically tied with the best LCB variant. It yields the strongest AURSAC on `HAM10000` and `Liver tumours`, while **Mean** edges out **Full** by a negligible margin on `Blood Cells` and in AURDAC on `Liver tumours`; these differences are well within one standard deviation. All LCB-based methods substantially outperform the non-deferring baseline in AURSAC.

Using LCB weights with $\alpha=1.0$ consistently matches or exceeds simpler alternatives on AURSAC$(0, 1.0)$, justifying **Full** as our default for uncertainty-aware deferral training.

### G.3 RQ3: Performance Scaling with Limited Context Data

We study how IFD and baselines exploit additional expert context at test time. During training, for each expert $E$ we compute the profile from a randomly drawn size-$n_c^E$ subset of its context $D_C^E$, *resampled each epoch*, which regularises the rejector and avoids overfitting to any single context realisation. On `Blood Cells` (Variable), we train with fixed small $n_c^E$ and evaluate as the context pool $N_C^E$ increases (15→1920).

Table 3: **Ablation of the LCB trust signal** (deferral budget $\tau \in [0,1]$). Values are mean $\pm$ sd over seeds; best *among LCB variants* in **bold**.

| Dataset | Metric | Full | | Mean | | Binary | | Classifier[†] | |
|---|---|---|---|---|---|---|---|---|---|
| | | Mean | SD | Mean | SD | Mean | SD | Mean | SD |
| Blood Cells | AURSAC ↑ | $0.668 \pm 0.029$ | | $0.669 \pm 0.027$ | | $0.665 \pm 0.028$ | | $0.595 \pm 0.025$ | |
| | AURDAC ↑ | $0.425 \pm 0.044$ | | $0.428 \pm 0.040$ | | $0.427 \pm 0.044$ | | $0.335 \pm 0.040$ | |
| HAM10000 | AURSAC ↑ | $0.641 \pm 0.042$ | | $0.639 \pm 0.040$ | | $0.638 \pm 0.044$ | | $0.603 \pm 0.014$ | |
| | AURDAC ↑ | $0.349 \pm 0.076$ | | $0.346 \pm 0.069$ | | $0.343 \pm 0.083$ | | $0.350 \pm 0.035$ | |
| Liver tumours | AURSAC ↑ | $0.634 \pm 0.018$ | | $0.632 \pm 0.021$ | | $0.628 \pm 0.027$ | | $0.562 \pm 0.018$ | |
| | AURDAC ↑ | $0.340 \pm 0.031$ | | $0.344 \pm 0.045$ | | $0.338 \pm 0.050$ | | $0.264 \pm 0.031$ | |

[†] AURDAC for **Classifier** is not meaningful because it never defers.

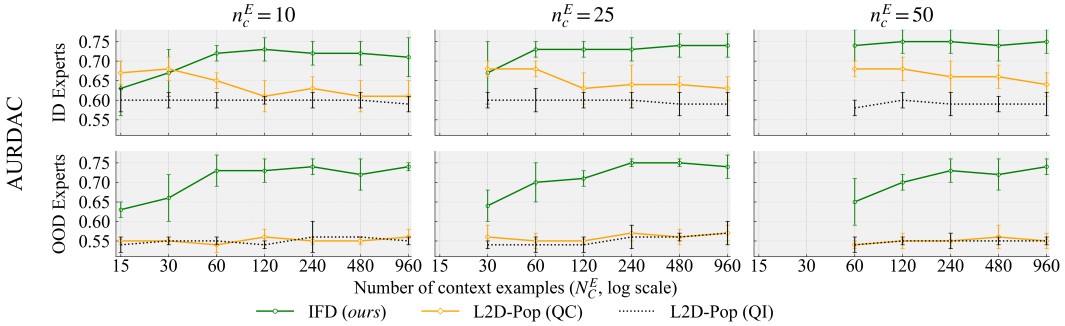

Figure 3: **Scaling with context.** `Blood Cells`, Variable Specialists. Models are trained with small per-expert subsamples $n_c^E \in \{10, 25, 50\}$ and evaluated as the available test-time context $N_C^E$ grows. IFD benefits from more context; L2D-Pop remains flat.

**Findings:** Figure 3 demonstrates that IFD consistently turns additional test-time context into better system accuracy and higher-quality deferrals, with gains emerging early and tapering at very large $N_C^E$. In contrast, L2D-Pop (QI) is largely insensitive to more context, and L2D-Pop (QC) tends to plateau or degrade as $N_C^E$ grows. These patterns hold across all training budgets ($n_c^E \in \{10, 25, 50\}$) and are most pronounced for OOD experts, reflecting that IFD's identity-free Bayesian profile naturally scales with context while fixed-dimensional, meta-trained encoders struggle to exploit large pools.

### G.4 ROBUSTNESS TO PRIOR MISSPECIFICATION

We evaluate a fixed, pre-trained IFD model on the `HAM10000` dataset without any retraining. All experiments vary only the evaluation-time priors and context; the classifier and rejector parameters remain fixed. We consider the Stable and Realistic experts defined in Section B. For each class $y$, a Beta prior $\mathrm{Beta}(\alpha_y, \beta_y)$ is parameterised by a mean $\mu_y$ and strength $S$, via $\alpha_y = \mu_y S$, $\beta_y = (1 - \mu_y)S$. We evaluate:

- Accurate Prior (AP): $\mu_y = \bar{\theta}_y^E$ (ground-truth profile).
- Uninformative Prior (UP): $\mathrm{Beta}(1, 1)$ (i.e., $S = 2$).
- Adversarial Prior (AdvP): "inverted" specialist (high competence on an incorrect class).
- Overconfident Generalist (AdvP–OCG): $\mu_y = 0.90$ for all $y$.

We sweep strong prior strengths $S \in \{10, 30, 100\}$ for AP, AdvP, and AdvP–OCG.

**Context Protocol** We vary the number of context points per class $n_C \in \{0, 5, 10, 20, 40, 80, 160, 320, 640\}$ (total context $N_C^E = K\, n_C$). For each $n_C > 0$, we sample context outcomes i.i.d. from the ground-truth expert profile $\bar{\theta}^E$. Posteriors are updated by conjugate Beta–Binomial updates per class.

**Evaluation** Given the posterior means and variances, the IFD rejector computes deferral probabilities for each expert on held-out test queries. We report the MAE of posterior means: $\frac{1}{K}\sum_{y=1}^{K}\left|\mu_y^E - \bar{\theta}_y^E\right|$.

**Aggregation** For each combination of expert archetype, prior scenario, strength $S$, and $n_C$, we perform $R = 20$ independent repeats (new context draws), and report mean $\pm$ standard error across repeats. Plots use $x$-axis $n_C^E$ and summarise MAE.

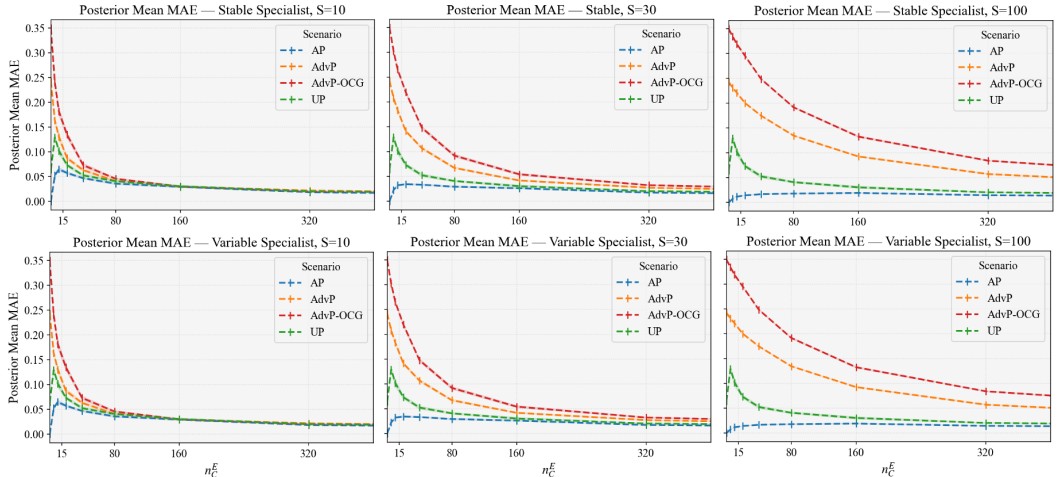

Figure 4: Adversarial priors experiment results. MAE vs context per class for Stable and Variable Specialists. Curves are near-identical; misspecified priors are overcome as context grows: $\approx 30$ points ($S = 10$), $\approx 320$ ($S = 30$), $\approx 640$ ($S = 100$).

**Findings:** Figure 4 displays the results. The trajectories are near-identical for Stable and Variable Specialists, indicating that sensitivity to prior misspecification is largely independent of the expert's polarity. MAE decreases monotonically with added context, and the context required to overcome a misspecified prior scales with prior strength $S$: for $S=10$, convergence occurs within $\sim 30$ context points; for $S=30$, around $\sim 320$; and for $S=100$, about $\sim 640$. These thresholds mark when the posterior effectively overrides the prior, as expected under Bayesian updating.

## H  DATASETS

We used three public medical imaging datasets to develop and test our methods. Each dataset is described below, noting its contents and license.

**HAM10000 Dataset**  The HAM10000 dataset (Tschandl et al., 2018) contains 10,015 dermatoscopic images of common pigmented skin lesions. It was created to provide a larger, more varied resource for training systems to identify skin conditions. Images were collected over twenty years from Vienna, Austria, and Queensland, Australia. The dataset covers seven diagnostic types: melanocytic nevi, melanoma, benign keratosis-like lesions, basal cell carcinoma, actinic keratoses, vascular lesions, and dermatofibroma. Over half the diagnoses were confirmed by histopathology, with the rest checked by follow-up, expert review, or confocal microscopy. The dataset is available via the Harvard Dataverse. The original paper states the license is Creative Commons Attribution 4.0 International (CC BY 4.0).

**Blood Cells Dataset**  The Blood Cells dataset (Acevedo et al., 2020), has 17,092 microscope images of single normal blood cells. It was made to help build and test systems for recognising blood cells, particularly using deep learning. The images are 360x363 pixel JPGs, taken with a CellaVision DM96 machine at the Hospital Clinic of Barcelona. Expert pathologists labelled the cells. The dataset groups cells into eight types: neutrophils, eosinophils, basophils, lymphocytes, monocytes, immature granulocytes, erythroblasts, and platelets. Immature granulocytes are grouped together because individually identifying them is less clinically important and difficult even for experts. This

grouping focuses on clearer categories for model training. The dataset is on Mendeley Data and uses the Creative Commons Attribution 4.0 International (CC BY 4.0) license.

**Liver tumours (LiTS) Dataset: Liver tumours**  The Liver tumour Segmentation (LiTS) dataset (Bilic et al., 2023) (referenced as "Liver tumours" in this paper) is a resource for testing liver and liver tumour segmentation methods. It includes 131 CT volumes for training and 70 for testing. Data came from seven hospitals and research centres, offering varied clinical cases. The dataset includes different tumour types, sizes, looks, and contrast levels, making it a tough, realistic test. The LiTS dataset and its online testing system are on CodaLab. It uses the Creative Commons Attribution-NonCommercial-NoDerivatives 4.0 International (CC BY-NC-ND 4.0) license.

