# OpenReview forum: "Identity-Free Deferral For Unseen Experts"
_ICLR.cc/2026/Conference — ICLR 2026 Poster_

### Official Review · Reviewer_z5Ji · 2025-10-25

**Soundness:** 4
**Presentation:** 3
**Contribution:** 4
**Rating:** 8
**Confidence:** 4

**Summary:**

This paper introduces Identity-Free Deferral (IFD), a new architecture for L2D systems that improves generalization to unseen human experts, especially those that are OOD to the training population. In standard L2D frameworks, a model learns when to predict and when to defer to an expert. Recent population-adaptive extensions condition this decision on learned expert embeddings, yet these architectures often rely on class-indexed signals that leak label identities and thus violate the permutation symmetry inherent in the problem. As a result, they tend to overfit to specific expert–class alignments and fail to transfer to unseen or re-indexed experts.

IFD addresses this issue by enforcing permutation invariance architecturally. Instead of embedding expert identities directly, it constructs a Bayesian, query-independent competence profile for each expert using a few-shot context of past predictions. The deferral rejector then operates on a compact, role-indexed state (containing structural information such as the model’s confidence in its top-ranked class and the expert’s estimated competence for that same role), thus eliminating all absolute class-identity channels. The model is trained with a novel uncertainty-aware, context-only loss that removes the need for query-time expert labels, weighting supervision by a lower confidence bound on expert reliability.

The authors provide formal proofs of permutation invariance, contrasting IFD with the non-invariance of standard population encoders, and show theoretically that the resulting policy focuses on structural relations between model and expert rather than identity-conditioned shortcuts. Empirical results across multiple medical imaging datasets and ImageNet-16H (with real human annotators) demonstrate that IFD consistently outperforms L2D-Pop and other baselines, especially for OOD experts and under input distribution shift. Moreover, IFD achieves these gains with substantially fewer expert annotations.

**Strengths:**

The paper tackles a highly relevant and practically important problem in L2D: ensuring reliable deferral to unseen or OOD experts. This setting reflects real-world scenarios such as hospital staff turnover, making the problem both timely and significant for AI deployment.

In terms of originality, the work offers a clear conceptual advance by identifying identity-conditioned shortcuts as a core limitation of existing L2D approaches and proposing an architectural solution (IFD) that enforces permutation symmetry by design. This symmetry-based perspective introduces a novel and principled inductive bias for expert adaptation.

The quality of the work is strong. The theoretical analysis is sound and well-motivated, including formal proofs of invariance and clear links to the Bayes-optimal rule. The accompanying uncertainty-aware, context-only training objective is practical, removing the need for costly query-time expert labels.

The experimental evaluation is thorough, spanning multiple datasets, both simulated and real experts, and a wide range of conditions such as in-distribution and OOD settings, input shifts, and ablations.

**Weaknesses:**

1) Although the theoretical analysis convincingly demonstrates that standard population-adaptive L2D architectures suffer from identity-conditioned shortcuts and that IFD enforces permutation invariance by design, the paper would benefit from a direct empirical illustration of these shortcuts. A simple toy experiment (e.g. showing how a conventional L2D-Pop model fails under systematic class reindexing while IFD remains stable) would make the argument more concrete and visually intuitive.

2) While the theoretical results appear sound, their scope could be clarified more explicitly. The proven invariance strictly holds under coherent relabelings (when both the model and expert are permuted consistently), whereas in practice the paper’s OOD setup often involves expert-only permutations where this assumption may not fully apply. Stating this distinction in the main text and briefly discussing potential extensions to partial or incoherent relabelings would improve conceptual precision.

3) The class-level abstraction of expert competence, though central to the method’s efficiency and invariance, inherently limits sensitivity to within-class, instance-specific expertise differences. While the authors acknowledge this in their discussion, a small-scale experiment or ablation (e.g., comparing performance when experts vary mainly within a class) would clarify how much this simplification affects performance.

4) There are minor presentation issues that should be addressed to improve polish and readability: the word “delerejector” in the abstract appears to be a typo, and Section 4.2 repeats the parenthetical phrase “(why this blocks identity leakage)".

**Questions:**

1) Could the authors include or describe a simple empirical toy example demonstrating this failure mode and how IFD avoids it?
2) The invariance proof assumes coherent relabelings (joint permutations of model, labels, and expert), while real OOD scenarios often involve expert-only shifts. Could the authors clarify how IFD behaves in such cases and whether approximate invariance can still be expected?
3) IFD models expert competence at the class level, ignoring within-class instance variation. How sensitive is performance to this assumption, and do the authors plan to explore instance-aware but still identity-free extensions?

---

> ### Author Response · Authors · 2025-11-17
>
> Thanks for your review.
>
> > minor presentation issues
>
> Thanks for highlighting this. We've corrected these, and uploaded a new PDF (changes in blue).
>
> **Questions:**
>
> **Question 1:**
> We thank the reviewer for this excellent suggestion. Following this feedback, we added a dedicated toy experiment}Appendix G.1 that directly visualizes the identity-conditioned failure mode of L2D-Pop and the invariance of IFD.
>
> In this 3-class setup, both models share the same feature extractor, seeds, and training configuration with the only difference in the competence summarization. When the expert’s class competence profile is permuted at test time (expert-only permutation), L2D-Pop fails to adapt, continuing to defer on the original class labels, while IFD correctly transfers deferral behavior to the new expert’s specialty. Figure 2 provides a clear visual comparison.
>
> **Question 2** This is an astute observation and an important clarification. Indeed, the invariance proof is w.r.t. coherent relabelings and the OOD problem setup is w.r.t. expert-only permutations. The purpose of proving invariance w.r.t. coherent relabelings is that it provides an easier and cleaner path to demonstrating why architectures with identity-channels can bias the deferral decision w.r.t. expert-only permutations.
>
> Concisely put, this is because coherent-relabeling invariance is an architectural, distribution-free property. If an encoder/rejector is invariant to every coherent relabeling then it cannot implement any function that depends on absolute label indices. Conversely, showing non-invariance proves the hypothesis class contains absolute-index (identity) dependent functions, which are the exact shortcuts that become brittle under expert-only reindexings.
>
> We do not directly prove the problem w.r.t. expert-only permutations because a direct proof that training will select such identity-shortcuts requires additional distributional and optimization assumptions (about the training data, empirical risk minimization, and optimizer dynamics); those are realistic but messy to state generally. The coherent-relabeling non-invariance is therefore a clean, general architectural diagnosis that (i) explains why identity channels permit the failure mode, and (ii) together with standard ERM/optimization behavior explains the empirical expert-only OOD failures we observe.
>
> You will see that we've changed the PDF to explicitly state this in the main text (lines ~234-253, ~526-533) and have briefly discussing potential extensions to partial or incoherent relabelings in the future works section. We've also tidied up §3.3 to improve clarity on this matter.
>
> **Question 3**: This is true. We think that the realistic expert simulation procedure (explained in App. B) used in our experiments and the findings in Table 1 answers this question. In short: the realistic expert simulation is based on correlated instance-dependent errors - not simple instance-independent errors used in prior papers. We determine two types of experts Stable and Variable, of which alter the level of instance-dependency. This protocol is designed to identify any fatal flaws in IFD's class-level expert competance, and provide a fair comparison to instance-dependent modelling such as L2D-Pop. Emprirically, from Table 2, we can draw the conclusions that IFD is not fatally sensitive to this assumption.
>
> We think exploring instance-aware but still identity-free extensions is the next natural step in this subfield of L2D. We leave this for open work to the community.

---

> > ### Comment · Reviewer_z5Ji · 2025-11-22
> >
> > Thank you for the detailed and thoughtful rebuttal. I appreciate the clarifications on the theoretical scope of the invariance arguments and, in particular, want to thank the authors for adding the toy experiment. This addition directly addresses my main suggestion and makes the core failure mode and the benefits of IFD much more concrete.
> >
> > Overall, the rebuttal satisfactorily resolves my questions and concerns. I continue to view the paper as a strong and meaningful contribution, and I am keeping my original score and recommendation for acceptance.

---

### Official Review · Reviewer_R4fz · 2025-10-29

**Soundness:** 3
**Presentation:** 2
**Contribution:** 3
**Rating:** 6
**Confidence:** 4

**Summary:**

The paper proposes Identity-Free Deferral (IFD) for Learning-to-Defer with population experts. The authors argue that common “population” encoders leak class/expert identity and are not invariant to coherent relabelings of classes, which can hurt transfer to new experts. Their approach build a small, permutation-invariant, role-indexed state from a query-independent competence profile estimated from a few context examples. The rejector compares the model’s confidence to an expert’s peak-class accuracy. Several experiments are made on different medical dataset, results are compared with the exist population L2D approach as well as a standard one.

**Strengths:**

- The work is sound and I understand the motivation on the symmetry.
- Reducing the number of predictions required by the expert is good.
- The method is quite practical and scalable.

**Weaknesses:**

-  I had a lot of trouble following the paper. The presentation could be substantially clarified: the structure feels dense, and it is often unclear how the different components are connected. A significant rewrite or reorganization might be needed to make the flow easier to follow.

- Some background and notation are imprecise or undefined. For example, $r(x)$ later becomes $r(x, E)$ without a clear definition, $\Delta$ is introduced but never defined, and LCB appears before Lower Confidence Bound is spelled out.

- Several small presentation issues and typos remain:
    - The 'QI' column is labeled [1].
    - Typo at line 10: 'delerejector',  line 1698: 'aggrerejector', parentheses line 229.

- The base surrogate loss employed in the paper has previously been shown to yield miscalibrated probability estimates [2] and to be non-realizable [3, 4].

- Only for a single expert.

**Questions:**

I have a couple of questions:

1. Your current surrogate loss appears largely inspired by [5]. Is there a specific reason why you compare primarily with the formulation from [1] rather than directly with that of [5]?

2. The base surrogate loss employed in the paper has previously been shown to produce *miscalibrated* probability estimates [2] and to be *non-realizable* under standard assumptions [3, 4]. Is there a particular reason you chose this formulation instead of those from [2] or [3], which are known to yield calibrated or realizable solutions?

3. In Table 3, you argue that **Full** consistently matches or exceeds other variants, but the difference between **Full** and **Mean** appears very small overall. Do you have an explanation for this? Is there a specific trade-off between $\mu_y^E$ and $\sigma_y^E$?

4. Out of curiosity, how does your approach perform in the standard Learning-to-Defer setting (with fixed experts), compared for instance to [1]?

5. Is there a reason why the current formulation is restricted to single expert?


------

[1] Rajeev Verma, Daniel Barrejon, and Eric Nalisnick. Learning to defer to multiple experts: Consistent surrogate losses, confidence calibration, and conformal ensembles.

[2] Yuzhou Cao, Hussein Mozannar, Lei Feng, Hongxin Wei, and Bo An. In defense of softmax parametrization for calibrated and consistent learning to defer.

[3] Anqi Mao, Mehryar Mohri, and Yutao Zhong. Realizable h-consistent and bayes-consistent loss
functions for learning to defer.

[4] Hussein Mozannar, Hunter Lang, Dennis Wei, Prasanna Sattigeri, Subhro Das, and David A. Sontag. Who should predict? exact algorithms for learning to defer to humans.

[5] Hussein Mozannar and David Sontag. Consistent estimators for learning to defer to an expert

---

> ### Author Response · Authors · 2025-11-17
>
> Thanks for your review.
>
> > presentation
>
> Parts of the paper are indeed compact, which is hard to avoid in a 9-page limit. However, with the extra page in the rebuttal we’re confident this can be greatly helped. In the newly uploaded PDF, you will see that we have addressed this across the paper, but primarily in §3.3 and throughout §4 (changes in blue). These come in the form of more in-depth descriptions and background of the purpose and contents of each subsection, providing greater cohesion and flow between sections.
>
> > notation/typos/background
>
> Thanks for pointing this out. We've corrected these in the newly uploaded PDF.
>
> > The base surrogate loss employed in the paper has previously been shown to yield miscalibrated probability estimates [2] and to be non-realizable [3, 4].
>
> The reviewer is correct about the base surrogate loss not being realizable $\mathcal{H}$-consistent. But that doesn’t undercut its legitimacy. It was still the first Bayes-consistent surrogate for L2D, and (importantly for practice) it admits $\mathcal{H}$-consistency bounds [6], which give finite-sample control of deferral risk via surrogate risk inside the chosen hypothesis class $\mathcal{H}$.
>
> Regarding calibration: prior work shows that the (K+1)-way softmax can be miscalibrated **when one reads the deferral posterior (or its odds) as the probability that the expert is correct**. IFD does not rely on that interpretation: the deferral decision is implemented by comparing the deferral logit to the top class logit, and our evaluation uses margin-based coverage/accuracy curves, of which neither requires calibrated probabilities of expert correctness. IFD’s rejector produces a deferral logit from an identity-free state, so it is naturally compatible with alternative calibrated parameterizations (e.g., OvA). We leave such substitutions as straightforward extensions.
>
> Additionally, experiments show the gaps between principled surrogates (realizable vs. not) are usually tiny (often sub-percentage and within one standard deviation). For instance, on HateSpeech the range is ~91.6–92.2 SA; on COMPASS it’s ~66.0–66.3, with methods trading off coverage more than raw system accuracy (see Table 2 of [3]).
>
> > Only for a single expert.
>
> This seems to be a misunderstanding. The formulation and experiments are designed for multi-expert throughout. In the original PDF, this was described as:
>
> *  §3 (``Population risk and Bayes rule'') defined the rejector as $r(x, E)$ and the population risk as an expectation over experts. The pop-Bayes rule is explicitly expert-conditioned. IFD inherits this as we compute an identity-free state $z(x, E)$ and a deferral logit $g⊥(x, E)$ for each expert.
> * Inference: Experiments paragraph "Evaluation under variable deferral budgets". Cases are ranked by a deferral margin, and the case is deferred to the expert with the largest deferral margin (i.e., winner-takes-all).
> * Table 1 report results across multiple experts with explicit number of ID/OOD experts.
>
> However, we do appreciate the reviewers feedback and perspective. This is an important aspect of the paper, so we have made changes to make the multi-expert aspect of our method abundantly explicit (lines 167-169 352-366 383-388 437-440). Hopefully this has cleared up this misunderstanding, please let us know otherwise.
>
> **Questions**:
> 1. (We think this is related to the single expert misunderstsanding). This is because [5] is suitable for single-experts only, whereas [1] handles multiple-experts.  Our framework (and experiments) are in the multi-expert setup.
> 2.   We deliberately kept the same base surrogate as L2D-Pop to primarly isolate the architectural issue we study. Swapping in a different surrogate would confound the comparison. Our goal is to show the effect of fixing L2D-Pop’s design while keeping the base loss the same.
> 3. This is because most experts had sufficiently large context sets, leading to small posterior variances. The $\textbf{Full}$ variant incorporates the lower-confidence bound term $(\mu - \alpha\sigma)$, which introduces an explicit trade-off between expected competence ($\mu$) and uncertainty ($\sigma$): higher $\sigma$ reduces the effective trust weight, discouraging deferral to uncertain experts. When context is large and $\sigma$ is small, this trade-off vanishes and $\textbf{Full}$ converges toward $\textbf{Mean}$. Under limited or noisy context, the $\sigma$ term provides useful regularization (prioritizing experts whose competence is both high and well-estimated) resulting in more stable deferral behavior.
> 4. (Again, we think this is related to the single-expert misunderstanding) Table 1 includes results for fixed-expert setups (final column). It does not perform as well as population-adaptive methods (as expected), but isn't far off.
>
> 5. Please see comment above re: "Only for a single expert."
>
> [6] Cross-Entropy Loss Functions: Theoretical Analysis and Applications, Mao et al., ICML 23

---

### Author Response · Authors · 2025-11-25
**quick update**

Hi reviewers. Just to let you know, we have slightly modified the PDF to include a sketch proof of why QI and QC encoders are generically non-invariant to coherent relabelings (following Thm 3.1 -- in blue). The full proof is still in the appendix, but we felt that including a brief sketch in the main body improves clarity and readability. Thanks

---

### Meta-Review · Area_Chair_bDov · 2026-01-02

**Summary:**

All reviewers are expected to be happy with this paper. One remaining common weakness is the paper representation – this should be further improved for general readers along with visuals.

**Reviewer Concerns:**

**Review R4fz**:
The reviewer raised six concerns and they are addressed as follows:
1. (**paper presentation**) I had a lot of trouble following the paper – the writing was slightly improved during the rebuttal.
2. (**clarification on the surrogate loss**) Concerns on the base surrogate loss, which yield miscalibrated probability estimates and to be non-realizable – addressed by the response saying that they are not required properties.
3. (**single expert**) Concern on the paper assumption on the number of experts – addressed as it is misunderstanding.
4. (**surrogate loss formulation**) The paper focuses on surrogate loss in [1] instead of [5]; why? – addressed as [1] is for multi-experts and this paper considers a multi-expert setup.
5. (**clarification on empirical results**) The reason that the difference between Full and Mean appears very small overall – explained (i.e., mainly due to the large context size).
6. (**comparison**) Performance under the standard Learning-to-Defer setting (with fixed experts) – addressed by empirically comparing in Table 1.

No outstanding concerns remained.


**Review z5Ji**:

1. (**motivational illustration**) It is beneficial to have a direct empirical illustration of limitations of identity-conditioned shortcuts – conducted additional experiments with illustration in Appendix G.
2. (**clarification on assumptions**) Is the assumption on expert-only permutations required? – clarified by confirming the expert-only permutations assumption.
3. (**performance sensitivity**) How sensitive is performance to the assumption of “expert competence at the class level”? – confirmed the limitation and explained in Table 1 and App. B.

No outstanding concerns remained.

**Reviewer Scores:**

**Review R4fz**:
Final expected rating: 6 / final expected confidence: 4 – I expect the reviewer will maintain scores or increase them as the most concerns are addressed.


**Review R4fz**:
Final expected rating: 8 / final expected confidence: 4 – The reviewer confirmed to maintain the original scores.

---

### Decision · Program_Chairs · 2026-01-26

Accept (Poster)